# POET: Partially Observed Earth Transformer with High-Dimensional Position Embedding

## Abstract

The Earth system is integral to every aspect of human life, and accurately forecasting the system states is vital in many domains. Current sensing technology can only obtain partial observations of the Earth, such as meteorological factors collected by multiple weather stations or flood monitoring in different river locations. In this paper, we focus on forecasting physical quantities into the future based on partial observations of scattered stations, recorded as high-dimensional time series. While Transformers are well-suited for processing 1D natural language or 2D vision data, their attention mechanism may struggle to learn higher-dimensional dependencies in Earth data. To advance data-driven Earth modeling, we present Partially Observed Earth Transformer, short as POET, which captures the 3D dependencies underlying the Earth system observations alternately from the temporal, spatial, and variate views. To tackle the position-insensitivity of the attention mechanism, we apply attention with a novel High-dimensional Position Embedding (HiPE) strategy that meticulously encodes the geographical bias of each Earth observation. HiPE not only effectively integrates the off-the-shelf prior knowledge into attention but also automatically discovers the latent relation in the high-dimensional system. In a set of empirical studies, POET achieves consistent state-of-the-art forecasting skills in weather, flood and air quality, across both global and regional Earth systems.

## 1 Introduction

The Earth system encompasses the dynamic and interconnected processes involving the atmosphere, hydrosphere, biosphere, and geosphere that sustain life on our planet (Steffen et al., 2005; Flato, 2011; Lenton, 2016). Understanding these complex interactions is essential for addressing global challenges such as climate change, resource management, and environmental sustainability (Shiroyama & Mino, 2011; Hornborg & Crumley, 2016; Steffen et al., 2018). In practice, we can access these complex spatiotemporal dynamics on the Earth through observation stations or sensors. However, due to equipment limitations, observations can only convey partial information about the complete system, necessitating the modeling of *partially observed Earth data* (Runge et al., 2019; Yu et al., 2024).

Unlike well-structured 1D language and 2D vision data, the Earth represents a complex system where an enormous number of variables interact intricately (Karpatne et al., 2018; Vance et al., 2024). Specifically, as illustrated in Figure 1, the Earth system observations form a *Multi-Station-Multi-Variate* framework (Wu et al., 2023), where each station is equipped with multiple sensors, designed to monitor a diverse range of environmental factors. Thus, these observations are inherently high-dimensional and multifaceted, which may encompass vast amounts of information across temporal, spatial, and variate dimensions, posing significant challenges in terms of representation learning and downstream analysis. In addition, the continuous and dynamic nature of the Earth system renders observations highly correlated. Taking the weather system as an example, geographically nearby stations may share similar meteorological environments and the observed physics quantities, e.g. pressure and temperature, are inherently interdependent. Therefore, how to effectively and efficiently capture *the multifaceted correlations underlying the high-dimensional Earth system* is the key to building a data-driven Earth model and promoting downstream applications.

Recently, deep learning models have achieved significant advances across a wide array of domains and tasks. Among these, Transformers (Vaswani et al., 2017) have garnered increasing attention over the past few years and become the major backbone of foundation models due to their capabilities of

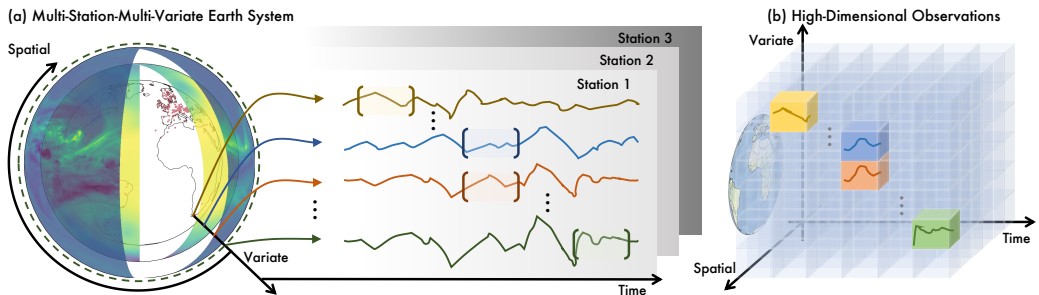

Figure 1: Partially observed Earth data are collected from multiple sensors located at multiple stations scattering across the globe. These observations are recorded as high-dimensional time series that lie in a three-dimensional data space, encompassing temporal, spatial and variate dimensions.

depicting pairwise dependencies empowered by the self-attention mechanism (Radford et al., 2018; Achiam et al., 2023). Originally introduced for natural language processing (Devlin et al., 2019), Transformers have become the dominant framework in computer vision (Dosovitskiy et al., 2020; Liu et al., 2021) and time series analysis (Nie et al., 2022; Liu et al., 2023). However, a notable limitation of the self-attention mechanism is its position-agnostic nature, which potentially overlooks crucial sequential and structural patterns inherent in the data. This drawback can be more severe in the high-dimensional data, the "first-class citizen" in the Earth system (Figure 1(b)), where the massive observation tokens may lead to degeneration of the attention mechanism. Although position embeddings are employed to incorporate position information into attention for ease of learning (Gehring et al., 2017; Dai et al., 2019), most research is predominantly limited to 1D natural language or 2D/3D vision data. How to identify, capture and utilize valuable position information in the context of high-dimensional time series observed from the Earth system remains a horizon to explore.

In this paper, we delve into the high-dimensional time series observed from the Earth system and identify the "position" of these observations in a three-dimensional data space, as illustrated in Figure 1(b). This physically structured organization benefits the disentanglement of massive observations and further inspires the **P**artially **O**bserved **E**arth **T**ransformer (**POET**), at the core of which is a cascaded attention mechanism that fully captures the 3D correlations underlying the temporal, spatial and variate dimensions. POET is boosted by a novel **Hi**gh-dimensional **P**ositional **E**mbedding (**HiPE**) strategy that meticulously encodes geographical bias in the form of both learned and prior position information. Empowered by HiPE, POET can effectively capture the geoscience-plausible 3D dependencies among large-scale high-dimensional observations. Experimentally, POET presents notable performance in several Earth modeling benchmarks. Our contributions are stated as follows:

- We propose POET as a general framework with a cascaded attention mechanism for partially observed Earth data, which can effectively capture the multifaceted interactions within the Earth system across the temporal, spatial, and variate dimensions.
- We introduce a high-dimensional position embedding tailored for the Earth system, namely HiPE, allowing POET to encode the prior positional information of numerous observations while discovering underlying correlations inherent in the data.
- We conduct extensive experiments on a wide range of Earth system forecasting benchmarks, covering weather, air quality, and flood forecasting. Experimentally, POET consistently achieves state-of-the-art performance with favorable interpretability.

## 2 RELATED WORK

### 2.1 EARTH SYSTEM FORECASTING

Earth system forecasting has long been recognized as a foundational challenge in science due to its pivotal role in understanding climate dynamics and predicting socioeconomic impacts (Reichstein et al., 2019; Steffen et al., 2020). Since dynamical systems are inherently tied to physical processes, traditional methods have been developed to simulate the interactions among various components of the Earth system based on theoretical principles and predefined PDE equations (Bauer et al., 2015). However, with the rapid proliferation of data collected from weather stations, radar, and satellites,

these physical models often struggle to fully utilize the abundance of available information, thereby hindering the ability to capture the complexity and variability inherent in real-world phenomena.

Deep learning methods have shown remarkable potential as efficient surrogate models for Earth system. Most existing approaches formulate the Earth system forecasting as a spatiotemporal forecasting problem. By accurately extrapolating from sequences of past radar maps, a substantial body of work has been developed to model spatiotemporal correlations. ConvLSTM (Shi et al., 2015) and PredRNN (Wang et al., 2022) integrate convolutional operations into the LSTM architecture to simultaneously capture spatial and temporal structures. With the rise of Transformers, FourCastNet (Pathak et al., 2022) combines the Vision Transformer (ViT) with Fourier-based token mixing to produce high-resolution forecasts, while Earthformer (Gao et al., 2022) employs a Cuboid Attention mechanism to capture diverse correlations by decomposing spatiotemporal tensors into cuboids through diverse cuboid decompositions. However, all of these methods rely on relatively dense observations of the region or even global spatiotemporal dynamics, whose input is expected to be in a regular grid. Thus, they are not applicable to the partially observed Earth data presented in this paper.

Recently, stations or sensors have become ubiquitous in the realm of Earth system due to their easy acquisition and deployment (Wu et al., 2023). However, these scattered observations are inherently governed by Earth dynamics that vary significantly across regions and time periods, posing substantial challenges for global forecasting. While deep learning models designed for time series forecasting excel at capturing temporal dynamics (Wu et al., 2022; Liu et al., 2024), they face difficulties in modeling correlations across stations. In contrast, POET is tailored to the partially observed Earth system, which can precisely capture intricate dependencies in the high-dimensional space.

## 2.2 POSITION EMBEDDING

Position embedding is an indispensable component of deep learning models, particularly for recent Transformer-based large language models (LLMs) (Raffel et al., 2020; Chowdhery et al., 2023). By incorporating absolute or relative position information, Transformers are enhanced with the capability to capture sequential relationships, especially in modeling long-context data (Touvron et al., 2023). In the original Transformer architecture (Vaswani et al., 2017), absolute positional embeddings are added to the input token embeddings, which can be trainable parameters or fixed sinusoidal functions designed to encode position indices in a continuous and interpretable manner. While effective, absolute positional embeddings have limitations in scenarios with varying-length or extremely long sequences. To address these drawbacks, relative position embeddings were introduced (Shaw et al., 2018; Dai et al., 2019), encoding positional offsets directly into the self-attention mechanism. These embeddings quantify the relative distance between the query and key tokens, reflecting the intuition that precise positional information becomes less relevant beyond a certain range.

Building on these advancements, Rotary Position Embedding (RoPE) (Su et al., 2024) has emerged as the dominant positional embedding technique in many large language model (LLM) designs. Technologically, RoPE encodes positional information through rotational transformations applied to the query and key vectors. This approach seamlessly incorporates relative positional relationships while preserving the expressivity and efficiency of the attention mechanism. Beyond its success in natural language processing, RoPE and its 2D extensions have been widely adopted in the vision domain, where they are usually used to encode spatial relationships in video data (Yang et al., 2024b; Wang et al., 2024a). In the field of time series analysis, canonical RoPE has been widely adopted (Shi et al., 2024; Liu et al., 2024; 2025) not only to address the permutation-invariance problem of self-attention but also to offer greater flexibility in handling long-context data.

## 2.3 LOCATION ENCODING

Location information serves as informative geospatial metadata in Earth system modeling (Mai et al., 2022). To effectively integrate this information, a substantial body of location encoding methods has been developed to embed geographic coordinates, e.g., longitude and latitude, into high-dimensional representations that facilitate spatial learning. As a representative, Wrap (Mac Aodha et al., 2019) normalizes the coordinates into spherical coordinates with sine and cosine functions to avoid discontinuities on the dateline. Afterwards, Space2Vec (Mai et al., 2020) introduces a multi-scale encoding framework that uses sinusoid functions with different frequencies to model absolute positions and spatial contexts. Rather than directly embedding raw coordinates, CARTESIAN3D

(Tseng et al., 2023) proposed to transform the position into 3D static in time Cartesian coordinates. Moving beyond Euclidean point distance modeling, Sphere2Vec (Mai et al., 2023) develops a unified view of distance-reserving encoding on spheres based on the Double Fourier Sphere. Recent SH (Rußwurm et al., 2024) propose to use orthogonal spherical harmonic basis functions paired with sinusoidal representation networks to learn representations of geographic location.

Building on location encoding, prior studies have primarily focused on processing geospatial image data, notably in satellite imagery (Cong et al., 2022; Rolf et al., 2024; Klemmer et al., 2025). Fewer efforts have been devoted to handling other data modalities, such as multivariate Earth-observed time series data studied in this paper. The high dimensionality of such data, spanning temporal, spatial, and variate dimensions, presents significant challenges for position encoding.

## 3 METHOD

To tackle the modeling difficulty of high-dimensional and highly correlated partially observed Earth data, we present POET with a high-dimensional position embedding strategy, which can effectively introduce valuable prior or latent knowledge vital for learning meaningful high-dimensional attentions.

**Problem formulation** In the problem of partially observed Earth system modeling, we are given the observations $\mathbf{x}_{1:T} = \{\mathbf{x}_1, \mathbf{x}_2, ...\mathbf{x}_T\}$, where $\mathbf{x}_t \in \mathbb{R}^{S \times V}$ denotes the data collected from all $S \times V$ sensors at time $t$. Here, $S$ is the number of stations, and $V$ is the number of physical variates collected in each station. Besides, the timestamp of each observation $\mathcal{T} = \{\mathcal{T}_i\}_{i=1}^T$ and geographic locations of all stations $\mathcal{G} = \{\mathcal{G}_j\}_{j=1}^S$ are also available to the deep model. In most cases $\mathcal{G}_j \in \mathbb{R}^2$ records the longitude and latitude for the $j$-th station. Notably, as shown in Figure 1, the position of a series of Earth observations is in a high-dimensional space, containing multifaceted position information. Such information can be incorporated into the model as prior knowledge to enhance the data-driven modeling of the Earth system. The goal of the Earth forecasting model $\mathcal{F}_\theta$ parameterized by $\theta$ is to predict the future $H$ timestamps based on the historical $T$ observations as well as the spatiotemporal prior information:

$$\widehat{\mathcal{X}}_{T+1:T+H} = \mathcal{F}_\theta \left( \mathcal{X}_{1:T} | \mathcal{T}, \mathcal{G} \right). \tag{1}$$

Following well-established time series modeling approaches, POET employs a patch-wise representation of observations to extract temporal information. Specifically, the input observation is divided into $N = \lfloor \frac{T}{P} \rfloor$ non-overlapping patches, where $P$ is the patch length. For the $i$-th patch of the observation from the $k$-th variate in the $j$-th station, denoted as $\mathbf{z}_{i,j,k}$, it is embedded into a $d_{\text{model}}$-dimensional token $\mathbf{h}_{i,j,k}$ through a trainable linear projection $\text{PatchEmbed}(\cdot) : \mathbb{R}^P \to \mathbb{R}^{d_{\text{model}}}$. Consider each series $\mathbf{x}_{:,j,k} \in \mathbb{R}^T$ of the $k$-th variate in the $j$-th station, its patching and embedding process writes as

$$\begin{aligned} \{\mathbf{z}_{1,j,k}, \mathbf{z}_{2,j,k}, \cdots, \mathbf{z}_{N,j,k}\} &= \text{Patchify} \left( \mathbf{x}_{:,j,k} \right), \\ \mathbf{h}_{i,j,k} &= \text{PatchEmbed} \left( \mathbf{z}_{i,j,k} \right), i = 1, 2, \cdots, N, \end{aligned} \tag{2}$$

where $\text{PatchEmbed}$ is a linear layer that projects the observation of $P$ consecutive time steps into a representation of $d_{\text{model}}$ channels. The high-dimensional time series $\mathbf{x}_{1:T}$ of the Earth system are embedded into a tensor $\mathbf{h}^0 = \{\mathbf{h}_{i,j,k}\} \in \mathbb{R}^{N \times S \times V \times d_{\text{model}}}$ and fed into the Transformer encoder.

### 3.1 HIGH-DIMENSIONAL POSITION EMBEDDING

Partially observed Earth data is organized into three orthogonal dimensions: temporal, spatial and variate. Given a series of observations, it can be simply located by a 3D coordinate $(i, j, k)$, corresponding to the step of time, the coordinate of space and the index of variate respectively. Intuitively, the time and space coordinates can be directly defined as the chronological and geographic information of the observation. However, we find that the absolute coordinates of temporal and spatial position can be misleading or fragile in many cases. For example, stations situated in close proximity but on opposite sides of a mountain might exhibit extremely different climate patterns and the distant timesteps still may present similar dynamics due to the periodicity. Besides, the variate coordinate is an integer index, too simple to fully reflect the complex relations among different physical variates. These thoughts motivate us to consider beyond prior information or data indices, which naturally leads to a High-dimensional Position Embedding (HiPE) strategy tailored to Earth observations.

**Position encoding**    Concretely, in HiPE, the position embedding comprises two types of position information: (1) *Prior position*, which refers to pre-existing information about the observation. In the context of Earth forecasting, such prior information typically corresponds to the spatial metadata of observation stations. (2) *Learnable position*, which introduces trainable positional components into the embedding, allowing the model to uncover hidden relationships within the observed data that may not be explicitly described by prior position. Based on the learnable position, the correlations between different variates can also be calculated by the relative distance in the learnable high-dimensional space. Technically, denoting temporal-, spatial-, and variate-position encodings of each Earth observation as $(p_i^{(t)}, p_j^{(s)}, p_k^{(v)})$, which are generated by

$$
\begin{aligned}
\text{Temporal} &: p_i^{(t)} = \mathcal{T}_i + \delta_i^{(t)}, i = 1, ..., N, \\
\text{Spatial} &: p_j^{(s)} = \mathcal{G}_j + \delta_j^{(s)}, j = 1, ..., S, \\
\text{Variate} &: p_k^{(v)} = \delta_k^{(v)}, k = 1, ..., V,
\end{aligned}
\tag{3}
$$

where $i, j, k$ are the position indices of the observation in the temporal, spatial and variate dimensions respectively. Notably, $\delta^{(t)} \in \mathbb{R}^{N \times 1}$, $\delta^{(s)} \in \mathbb{R}^{S \times 2}$, $\delta^{(v)} \in \mathbb{R}^{V \times C}$ are learnable parameters to capture data-informed position bias. Given the intricate multivariate relationships, the learnable position in the variate dimension is in high-dimensional space where $C$ is a hyperparameter.

**Rotary position embedding**    In HiPE, as presented in Figure 2, we extend the advanced RoPE (Su et al., 2024) technique to directly integrate the position encoding into the attention mechanism inside the model. Specifically, in RoPE, the representation of $N$ query and key tokens of the attention mechanism will be multiplied by a rotation matrix for $\{p_i\}_{i=1}^{N}$ degrees. Benefiting from the rotation matrix property, the dot product of the $i$-th query token and the $j$-th key token will be weighted according to the intersection angle between $p_i$ and $p_j$, successfully introducing the position information by reweighting the dot-product attention.

In contrast to the traditional 1D RoPE that is limited to encoding one-dimensional position information, HiPE needs to embed the position encoding (Eq. 3) of more than one dimension. Thus, we propose to ascribe the learned multifaceted position to different subspaces of the learned representations. Specifically, given position information of $C$ dimensions, we divide the hidden representation into $C$ subspaces along the channel dimension and independently apply 1D RoPE on each subspace with the corresponding dimension of the position information. The resulting encoded representations are then concatenated along the channel dimension to obtain the final rotary position embedding. Therefore, given the query or key representation $\mathbf{h} \in \mathbb{R}^{d_{\text{model}}}$ of each token and its corresponding position information $p \in \mathbb{R}^C$, the rotary position embedding process can be formalized as follows:

$$
\text{HiPE}(\mathbf{h}, p) = \text{Concat}\left([\mathbf{R}_{p_1}\mathbf{h}_1, \mathbf{R}_{p_2}\mathbf{h}_2, \ldots, \mathbf{R}_{p_C}\mathbf{h}_C]\right), 1 \leq c \leq C.
\tag{4}
$$

Here $\mathbf{h}_c \in \mathbb{R}^{\frac{d_{\text{model}}}{C}}$ denotes the split hidden representation of $\mathbf{h}$, and $\mathbf{R}_{p_c}$ is corresponding rotation matrix for the $c$-th dimension of position $p$. Subsequently, HiPE will be applied to queries and keys in the attention mechanism, which will be introduced in the next section.

## 3.2    PARTIALLY OBSERVED EARTH TRANSFORMER

As shown in Figure 2, POET adopts an encoder-only transformer architecture, consisting of three self-attention layers to capture temporal, spatial, and variate dependencies, respectively. Notably, these three orthogonal attention mechanisms, which can be flexibly arranged in any order, are further enhanced with HiPE by incorporating prior and learned "position" information.

**Earth Transformer Encoder**    To address the high-dimensionality of Earth observation, POET comprises stacked attention mechanism to capture the dependencies within the Earth system across three distinct dimensions. Specifically, for each Earth Transformer block, we employ separated temporal, spatial and variate attention. Instead of using the permutation-invariant self-attention mechanism, POET incorporates the explicit, dimension-specific position embedding with a rotation matrix to enhance POET with position awareness. Suppose there are $L$ layers, the $l$-th layer of the

Figure 2: Overall design of POET, which is an encoder-only model, comprising attention layers from three dimensions: temporal, spatial, and variate. High-Dimensional Position Embedding, containing an absolute position derived from prior position information and a learnable position, is incorporated in the formulation of self-attention to introduce underlying latent relations in high-dimensional space.

Earth Transformer Encoder can be formalized as follows:

$$\text{Temporal: } \hat{\mathbf{h}}^{l,t} = \text{LayerNorm}\left(\mathbf{h}^{l-1} + \text{Attn}\left(\text{HiPE}(\mathbf{q}^{l-1}, p^{(t)}), \text{HiPE}(\mathbf{k}^{l-1}, p^{(t)}), \mathbf{v}^{l-1}\right)\right),$$

$$\text{Spatial: } \hat{\mathbf{h}}^{l,s} = \text{LayerNorm}\left(\hat{\mathbf{h}}^{l,t} + \text{Attn}\left(\text{HiPE}(\hat{\mathbf{q}}^{l,t}, p^{(s)}), \text{HiPE}(\hat{\mathbf{k}}^{l,t}, p^{(s)}), \hat{\mathbf{v}}^{l,t}\right)\right), \quad (5)$$

$$\text{Variate: } \hat{\mathbf{h}}^{l,v} = \text{LayerNorm}\left(\hat{\mathbf{h}}^{l,s} + \text{Attn}\left(\text{HiPE}(\hat{\mathbf{q}}^{l,s}, p^{(v)}), \text{HiPE}(\hat{\mathbf{k}}^{l,s}, p^{(v)}), \hat{\mathbf{v}}^{l,s}\right)\right),$$

where $\mathbf{q}^*, \mathbf{k}^*, \mathbf{v}^*$ are projected from the hidden representation $\mathbf{h}^*$ with linear layers. $\mathbf{h}^l$ is the output of the $l$-th layer, $l \in \{1, ..., L\}$. Here, $\text{Attn}(\cdot)$ indicates attention mechanism enhanced with high-dimensional position embeddings derived from Eq. 4, where $p^{(t)}, p^{(s)}, p^{(v)}$ denote the position information for the temporal, spatial and variate dimensions, respectively. By alternately applying attention across these dimensions, the dimension-decoupled design can enable a comprehensive interaction among the three dimensions of Earth observation in a computationally efficient way.

**Collaborative Forecasting**    Finally, we adopt a shared linear regressor along the temporal dimension of $\mathbf{h}^L$ to generate the final prediction, enabling collaborative forecasting for all the observations. L2 loss between the prediction and the ground truth is adopted as the objective function for training.

## 4   EXPERIMENTS

To verify the effectiveness and generality of our proposed POET, we conduct extensive experiments under three challenging real-world Earth-observed benchmarks, ranging from meteorological indicators, air quality, and river discharge. For all datasets, the prior position in spatial dimension is the geographic coordinate of each station and the dimension of the learnable variate position is fixed at $C = 3$. Ablation on the dimensionality is presented in Table 8 in the Appendix.

### 4.1   METEOROLOGICAL FORECASTING

**Setups**    Meteorological forecasting is an everlasting problem in Earth system modeling as it poses significant challenges in learning complex temporal dynamics and variable correlations. In this section, we explore the effectiveness of our proposed approach on the Global Temperature and Wind Speed Forecasting challenge benchmark (GTWSF) (Liu et al., 2024), which contains the hourly averaged wind speed and hourly temperature of 3850 stations around the world spanning two years. The objective of the forecasting task is to predict the indicators for the next day based on the past two days' data, where the input length is 48 hours and the forecast length is 24 hours. Since these two benchmarks are derived from the same weather stations, we combined them together and trained a unified model to further demonstrate the efficacy of our design.

**Results**    As shown in Figure 3, POET demonstrates remarkable performance on two meteorological forecasting benchmarks, comprehensively surpassing classic statistical-based methods and recent

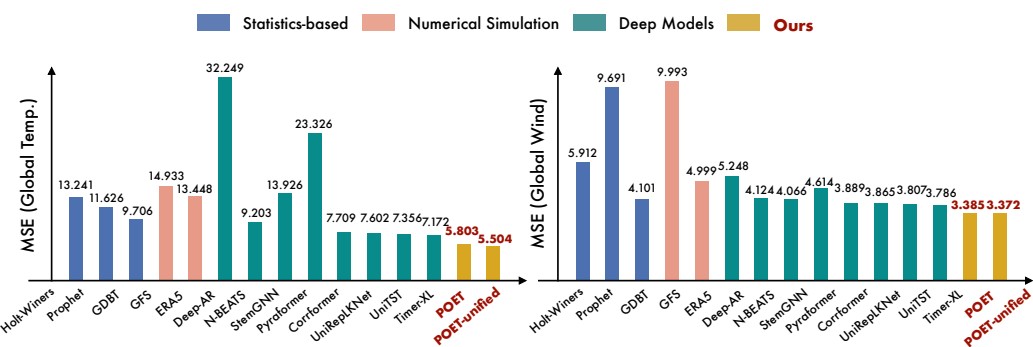

Figure 3: Forecasting results of two meteorological observations, collected from 3850 worldwide stations spanning two years. Results of the baseline models are sourced from Timer-XL (2024).

advanced deep models by a large margin. Surprisingly, POET-unified, trained on the combined datasets from these benchmarks and evaluated separately, achieves substantial improvements over the state-of-the-art multivariate time series forecasting model Timer-XL (Liu et al., 2024), with MSE reductions of **19.09%** and **10.59%** on global temperature and wind benchmarks, respectively. Moreover, the unified model achieves relative improvements of 5.15% and 0.38% compared to the standalone POET model. These results underscore the effectiveness of variate correlations in Earth system modeling. Additionally, even when trained on a single benchmark, POET consistently outperforms Timer-XL, which uses multivariate series as sequences and employs advanced positional embeddings to model complex dynamics and correlations. The significant improvements highlight the critical role of spatial position information in forecasting real-world observational data.

## 4.2 AIR QUALITY FORECASTING

**Setups** In this section, we conduct extensive experiments on a real-world Air Quality forecasting benchmark (Hettige et al., 2024), which is a multi-station-multi-variate dataset, recording air quality and weather-related observations from 35 major monitoring stations in Beijing. Following prior studies, we forecast PM2.5 concentrations for the next 72 hours (24 steps) using data from the preceding 72 hours and evaluate forecasting performance across different lead times: 24 hours (8 steps), 48 hours (16 steps), and 72 hours (24 steps). Unlike previous works (Wang et al., 2020; Liang et al., 2023) that focus solely on a single target variable, we extend the evaluation to encompass multiple observations and compare POET with advanced multivariate forecasting baselines.

Table 1: Performance on Air Quality forecasting. The look-back horizon is set to 72H. A lower MAE indicates a better prediction. Results of baselines are officially reported by (Hettige et al., 2024).

| Type | Classic Methods | | Neural DE Network | | Deep Spatio-temporal Models | | | | | | | |
|---|---|---|---|---|---|---|---|---|---|---|---|---|
| Model | HA | VAR | LatentODE | ODE-LSTM | DCRNN | STGCN | GMAN | GTS | PM25GNN | AirFormer | AirPhyNet | **POET** |
| **24H** | 38.37 | 60.10 | 44.83 | 46.19 | 35.99 | 33.70 | 50.62 | 34.99 | 50.94 | 29.62 | 29.11 | **23.19** |
| **48H** | 45.80 | 60.44 | 45.95 | 49.18 | 49.66 | 38.93 | 50.73 | 54.18 | 48.81 | 38.43 | 36.69 | **29.30** |
| **72H** | 50.58 | 60.64 | 47.14 | 51.45 | 57.01 | 43.93 | 50.69 | 73.50 | 51.51 | 43.39 | 42.23 | **33.53** |
| **AVG** | 44.92 | 60.39 | 45.97 | 48.94 | 47.55 | 38.86 | 50.68 | 54.22 | 50.42 | 37.15 | 36.01 | **28.68** |

**Results** Results in Table 1 demonstrate that POET achieves consistent state-of-the-art performance. Among classic methods, Neural DE Networks, and advanced deep spatio-temporal models, POET performs best across different forecasting horizons. Given that the air quality dataset comprises multiple observation indicators from each station, we further evaluate forecasting performance across all indicators and compare it with the current state-of-the-art multivariate forecasting models. As detailed in Table 11 in the Appendix, POET demonstrates superior collaborative forecasting capability, highlighting its versatility and generalizability in handling various meteorological indicators.

## 4.3 RIVER DISCHARGE FORECASTING

**Setups** In this section, we further conduct experiments on the recently proposed hydrology benchmark CausalRivers (Stein et al., 2025), which contains three diverse datasets, recording river discharge

Table 2: Forecasting performance on the CausalRivers benchmark. We report MSE results here, where a lower value indicates better prediction. MAE results are listed in Table 12 in the Appendix. We follow the standard protocol of long-term forecasting (Wang et al., 2024b), where both the input length and prediction length are set to 96 for all baselines. Avg means the average results from all three datasets: Flood, Germany and Bavaria. "-" denotes the out-of-memory (OOM) problem.

| Model | Autoformer | SCINet | DLinear | TimesNet | TiDE | Crossformer | PatchTST | iTransformer | TimeXer | **POET** |
|---|---|---|---|---|---|---|---|---|---|---|
| Flood | 0.161 | 0.078 | 0.096 | 0.081 | 0.077 | 0.095 | 0.068 | 0.070 | 0.066 | **0.064** |
| Germany | 0.381 | 0.312 | 0.235 | 0.294 | 0.242 | 0.240 | 0.229 | 0.235 | 0.236 | **0.225** |
| Bavaria | 1.660 | 0.434 | 0.383 | - | - | 0.344 | 0.345 | 0.344 | 0.335 | **0.323** |
| AVG | 0.734 | 0.275 | 0.238 | - | - | 0.226 | 0.214 | 0.217 | 0.212 | **0.204** |

from multiple observation stations within a specific area at a 15-minute temporal resolution. Since the benchmark was originally introduced for causal discovery, we adhere to the prevalent long-term time series forecasting protocol to predict river discharge for one day (96 timestamps) based on observations from the previous day. We thoroughly include well-acknowledged and advanced forecasting models as our baseline models, the implementation details are listed in Appendix A.2.

**Results**  As shown in Table 2, POET consistently outperforms other advanced deep forecasting models. Essentially, Transformers that are designed to capture complex temporal variations or variate correlations, reasonably demonstrate strong performance. It is also notable that the baseline model, TimeXer, employs two separate attention mechanisms to capture dependencies across the temporal and variable dimensions, achieving the second-best average performance across all three datasets. In comparison, POET enhances the position-insensitive attention mechanism by integrating multifaceted positional information for each dimension, thereby achieving significant advancements in forecasting performance and further underscoring the effectiveness of the proposed HiPE.

## 4.4 MODEL ANALYSIS

**Ablation Studies**  In addition to the main results, we also conduct comprehensive ablation studies to verify the effectiveness of our proposed POET, covering both the POET encoder design and the high-dimensional position embedding components. Specifically, for the architectural design, we remove the temporal, spatial, and variate attention layers, respectively. Results in Figure 7 in the Appendix demonstrate that all three types of attention are favorable for the prediction. To further validate the high-dimensional embedding components, we retain the existing model architecture and remove the prior positional embedding and the learnable embedding, respectively. Additionally, we compare the performance of HiPE with the canonical RoPE, which was originally designed for language models. RoPE operates on one-dimensional positional information and is limited to application within the temporal dimension. The results are listed in Table 3, where POET demonstrates superior performance across all datasets, outperforming all other ablations.

**Analysis of Position Embedding**  In POET, positional information is incorporated into time series modeling during the attention formulation. We further conduct a comprehensive analysis of its effectiveness against existing location encoding methods. Technologically, we maintain the Transformer design of POET while replacing the RoPE mechanism in the attention layers with a learnable node-

Table 3: Ablation results on the design of High Dimensional Position Embedding. *Prior.* and *Learn.* are abbreviations for the prior position and learnable position respectively. *Temp RoPE* represents applying the canonical rotary position embedding in the temporal dimension.

| Design | | Flood | | Germany | | Bavaria | | Wind | | Temp | | AVG | |
|---|---|---|---|---|---|---|---|---|---|---|---|---|---|
| | | MSE | MAE | MSE | MAE | MSE | MAE | MSE | MAE | MSE | MAE | MSE | MAE |
| W/ HiPE | W/o Prior | 0.068 | 0.118 | 0.228 | 0.155 | 0.329 | 0.166 | 3.527 | 0.127 | 6.135 | 0.167 | 2.058 | 0.147 |
| | W/o Learned | 0.068 | 0.118 | 0.230 | 0.156 | 0.331 | 0.168 | 3.471 | 0.126 | 6.046 | 0.165 | 2.029 | 0.146 |
| W/o HiPE | W/o HiPE | 0.068 | 0.117 | 0.228 | 0.155 | 0.339 | 0.168 | 3.878 | 0.134 | 7.343 | 0.184 | 2.371 | 0.152 |
| | W/ Temp RoPE | 0.070 | 0.120 | 0.232 | 0.156 | 0.341 | 0.169 | 3.882 | 0.134 | 7.297 | 0.183 | 2.364 | 0.152 |
| **POET** | | **0.064** | **0.115** | **0.225** | **0.155** | **0.323** | **0.163** | **3.385** | **0.125** | **5.803** | **0.162** | **1.960** | **0.144** |

Table 4: Comparison of POET with other position encoding methods. 3D refers to CARTESIAN3D. We follow the location encoding baselines used in (Rußwurm et al., 2024). Results on Air Quality are listed in Table 6 in the Appendix.

| Method | w/oPE | Node | Naive | Direct | 3D | RBF | RFF | Theory | Wrap | Sphere | SH | **POET** |
|---|---|---|---|---|---|---|---|---|---|---|---|---|
| Wind | 3.878 | 3.767 | 3.510 | 3.893 | 3.895 | 3.868 | 3.520 | 3.854 | 3.896 | 3.515 | 3.938 | **3.385** |
| Temp | 7.343 | 7.105 | 6.924 | 7.574 | 7.762 | 7.289 | 6.132 | 7.926 | 7.917 | 6.449 | 8.376 | **5.804** |

specific embedding method (Cini et al., 2023) and several location encoding modules (Mac Aodha et al., 2019; Mai et al., 2020; 2023; Rußwurm et al., 2024). Results in Table 4 reveal that position embedding methods generally outperform the model without positional encoding, highlighting the need for spatial information in Earth system modeling. Notably, the proposed POET consistently outperforms the alternatives.

**Analysis of Variate Correlations**   We conduct a comprehensive visualization analysis of POET to further validate its interpretability, particularly concerning variate correlations. In our proposed HiPE, we employ a data-driven learnable position embedding to capture the dataset-level variate correlations. Since there is no prior information about the relationships among variates at this level, the learned relative position can be viewed as a dataset-level representations of variate interdependencies, which in turn enhance the model's interpretability. To illustrate this, we visualize both the learned attention map and the learned position on each token. As shown in Figure 4 (Left), the learned relative positional distances effectively reflect the correlations among different variates, leading to a distinguishable attention map. For instance, when focusing on the variable Wind Speed, the variables with the closest and farthest relative positions are pressure and wind humidity, respectively, which aligns with meteorological principles and domain knowledge.

**Analysis of Spatial Proximity**   Although absolute two-dimensional latitude and longitude coordinates of the stations are known to the model, we also introduce learnable positional embeddings in the spatial dimension to incorporate data-driven information and enhance the modeling of spatial proximity. To validate this design, we conduct a comprehensive analysis on the GTWSF benchmark, which includes 3,850 stations worldwide. Figure 4 illustrates the position shift of stations situated along the eastern coast of the United States after integrating the learnable spatial embeddings. Notably, with the inclusion of the learnable position, all coastal stations exhibit a similar pattern of shift, predominantly moving towards inland. Since there are no weather stations located in the ocean, coastal stations in the dataset are more likely to exhibit stronger learned correlations with nearby inland stations than those separated by the sea. Beyond encoding the original geographic information, our proposed POET effectively captures these latent spatial relationships and dependencies, offering a comprehensive understanding of the underlying correlations.

**Case Study**   We present a case study on the forecasting performance of POET using the GTWSF benchmark at two stations in China, as shown in Figure 5. Specifically, we visualize the prediction results for each station. Notably, there are clear differences in the temporal variation patterns and

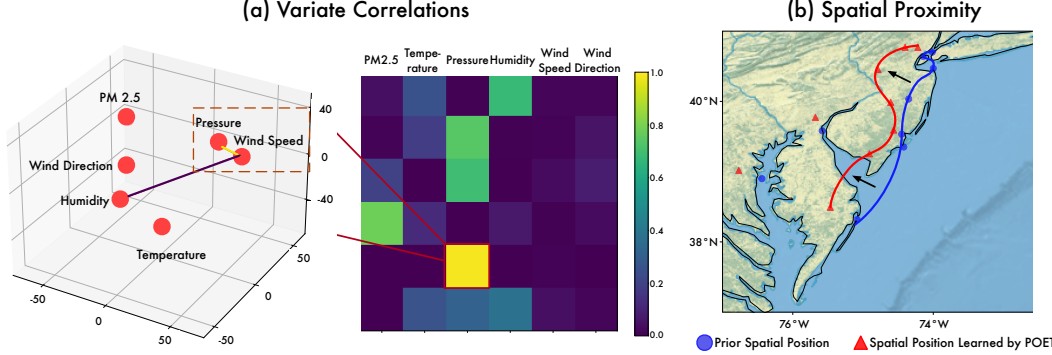

Figure 4: Model analysis on the learned Variate Correlations and Spatial Proximity. (a) Visualization of learned 3D position for each variate and corresponding attention maps on the Air Quality benchmark. (b) Comparison between prior spatial position and spatial position learned by POET on the GTWSF datasets, focusing on stations located along the eastern coast of the United States.

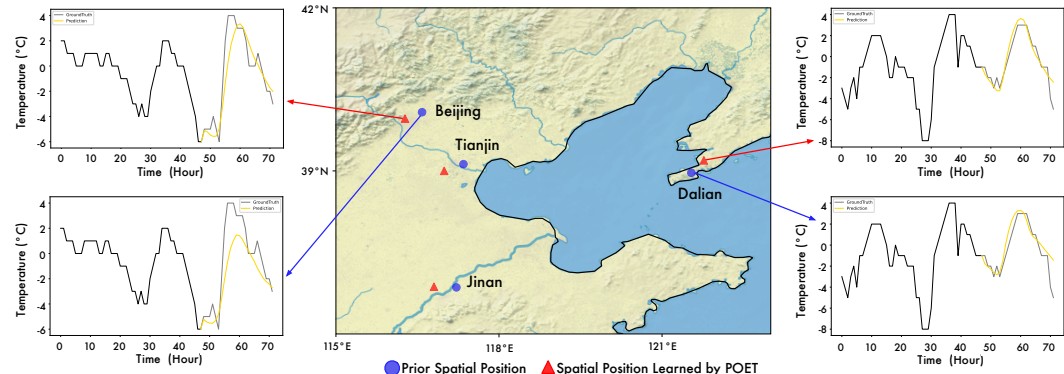

Figure 5: Forecasting case of temperature from 2020/12/03 18:00 to 2020/12/04 18:00 at Beijing (40°08'N, 116°58'E) and Dalian (38°97'N, 121°54'E).

numerical ranges between the two stations. Therefore, the relative distance between the spatial positions learned by POET is significantly greater than their prior geographical positions, reflecting weaker dependencies between these two stations. By incorporating this learned spatial proximity, POET achieves superior performance compared to using only absolute geographical information.

## 5 CONCLUSION

In this paper, we introduce POET, a novel Transformer-based architecture for Earth system forecasting. Toward high-dimensional Earth observation, we propose a high-dimensional positional embedding method termed HiPE. HiPE introduces deft positional information for temporal causality, variate correlation, and spatial proximity through a combination of prior information and learnable embeddings. By integrating high-dimensional positional information into the attention mechanism of each corresponding dimension, POET is enhanced with the capability to capture the underlying intricate dependencies among numerous observations with favorable interpretability. Experimentally, POET achieves significant advancement across diverse Earth observation forecasting scenarios.

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

# A  EXPERIMENTAL DETAILS

## A.1  DATASETS

We conduct experiments on well-acknowledged real-world datasets to evaluate the performance of the proposed POET, which include (1) **GTWSF** (Liu et al., 2024), collected by the National Centers for Environmental Information (NCEI), offers hourly wind speed and temperature records from 3,850 globally distributed meteorological stations spanning 2019-2020, enabling cross-scale weather forecasting and climate research through its global network. (2) **Air Quality** dataset (Hettige et al., 2024) includes hourly air quality and meteorological data spanning January 1, 2017, to May 30, 2018, collected from 35 major monitoring stations in Beijing. The dataset contains six variables, including the concentration of a pollutant (PM2.5) and five weather attributes (temperature, barometric pressure, humidity, wind speed, and wind direction), which are recorded every 3 hours. (3) The **CausalRivers** (Stein et al., 2025) benchmark dataset is constructed based on high-frequency observations of river flows from hydrological monitoring stations in multiple states of Germany, with a core covering continuous monitoring records with 15-minute accuracy from 2019 to 2023. The train/val/test split ratio is 7:1:2 for all datasets. The dataset contains three specialized subsets:

- **RiversEastGermany**: RiversEastGermany covers 666 monitoring stations in six eastern German states, encompassing a wide variety of hydrological environments such as plain rivers, mountain streams, and man-made canals in the Berlin metropolitan area, with a high degree of time-series completeness.

- **RiversBavaria**: RiversBavaria focuses on 494 stations in Bavaria, covering transboundary water systems such as the Danube and Main rivers as well as glacial meltwater areas in the Alpine foothills. The subset shows significant elevation gradients and seasonal flow fluctuations.

- **RiverElbeFlood**: RiverElbeFlood captures ultra-dense observations from 42 stations during pre-disaster warnings, flood evolution, and recession for the 2023 extreme flood event in the Elbe River Basin. This subset contains records of dam failure events at 6 hydrologic stations, showing strong distributional shifts.

We further visualized the locations of all the stations in each dataset on the map, and the results are presented in Figure 6.

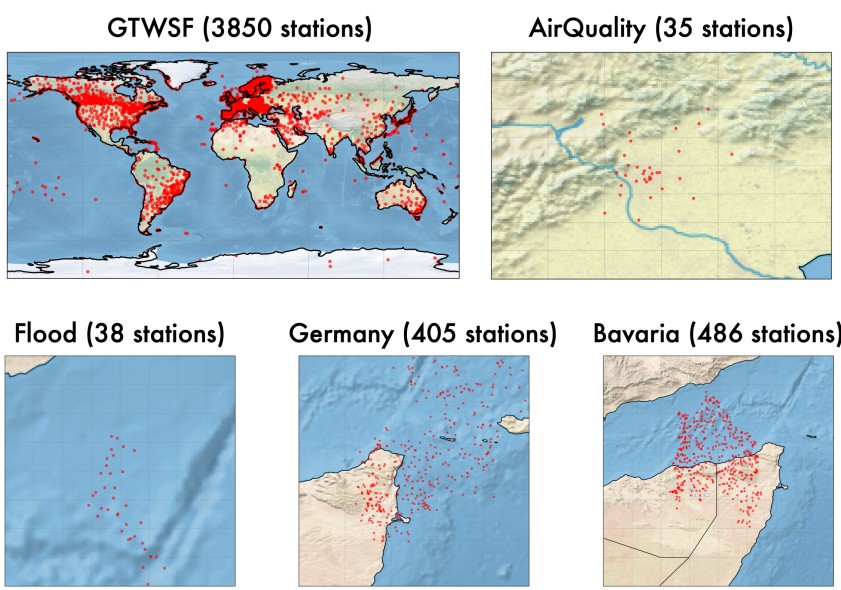

Figure 6: Visualization of stations distribution in each benchmark.

## A.2 BASELINES

We aim to present POET as a foundation model for high-dimensional time series forecasting. We thoroughly include well-acknowledged and advanced models in each forecasting task. For the GTWSF dataset, we report the official results of Timer-XL (Liu et al., 2024). For the Air Quality dataset, we not only compared the results from the AirPhyNet (Hettige et al., 2024), which includes two classic methods, two Neural DE Networks, and seven Deep Spatio-temporal Models but additionally with the advanced models. For the CausalRivers dataset, we compare POET with TimeXer (Wang et al., 2024c), iTransformer (Liu et al., 2023), PatchTST (Nie et al., 2022), Crossformer (Zhang & Yan, 2023), TiDE (Das et al., 2023), TimesNet (Wu et al., 2022), DLinear (Zeng et al., 2023), SCINet (Liu et al., 2022), and Autoformer (Wu et al., 2021). Totally, more than twenty baselines are included for a comprehensive comparison.

## A.3 IMPLEMENTATION DETAILS

All the experiments are implemented by PyTorch (Paszke, 2019) and conducted on NVIDIA 4090 24GB GPU. We utilize ADAM (Kingma & Ba, 2014) with an initial learning rate $10^{-4}$ and MSE loss for model optimization. The training precess is fixed to 10 epochs with an early stopping. We set the number of Transformer blocks in our proposed model $L \in \{1, 2, 3\}$. The dimension of series representations $d_{model}$ is set from $\{128, 512, 1024\}$. The patch length is fixed depending on the dataset. The patch length defaults to 24 for GTWSF and CausalRivers benchmarks, and which is set to 4 in Air Quality due to the limited input length. Partially, we reproduced the compared baseline models based on Time-Series-Library (Wang et al., 2024b). The results of other baselines are based on the benchmark provided by Timer-XL; AirPhyNet, which is fairly built on the configurations provided by their original paper. We provide detailed experimental configurations in Table 5.

To evaluate model performance, we utilize the Mean Squared Error (MSE) and Mean Absolute Error (MAE) metrics, consistent with established methodologies in prior research. These metrics are mathematically defined as follows:

$$\text{MSE} = \sum_{i=1}^{T} |\mathbf{X}_i - \widehat{\mathbf{X}}_i|^2, \quad \text{MAE} = \sum_{i=1}^{T} |\mathbf{X}_i - \widehat{\mathbf{X}}_i|.$$

Here $\mathbf{X} \in \mathbb{R}^T$ is a univariate time series and $\widehat{\mathbf{X}}$ is the corresponding prediction. For multi-station-multi-variate time series, the metrics are aggregated by averaging across both the station and variable dimensions to ensure a comprehensive evaluation.

Table 5: Experimental configurations of POET. All the experiments adopt the ADAM (Kingma & Ba, 2014) optimizer with the default hyperparameter $(\beta_1, \beta_2) = (0.9, 0.999)$.

| Dataset | Configuration | | | | | Training Process | | | |
|---|---|---|---|---|---|---|---|---|---|
| | $L$ | $D$ | $d_k$ | $H$ | $P$ | LR | Loss | Batch Size | Epochs |
| Flood | 1 | 512 | 64 | 8 | 96 | 0.0001 | MSE | 32 | 10 |
| Germany | 1 | 512 | 64 | 8 | 96 | 0.0001 | MSE | 64 | 10 |
| Bavaria | 1 | 1024 | 128 | 8 | 96 | 0.0001 | MSE | 32 | 10 |
| Wind | 3 | 1024 | 128 | 8 | 96 | 0.0001 | MSE | 4 | 10 |
| Temp | 3 | 1024 | 128 | 8 | 96 | 0.0001 | MSE | 4 | 10 |
| Air Quality | 1 | 512 | 64 | 8 | 96 | 0.0001 | MSE | 4 | 10 |

# B ABLATION STUDY

**Analysis of Position Embedding** In the main text, we conduct a comprehensive comparison between POET and other location encoding methods on GTWSF datasets. To further validate its superiority on regional data, we conduct experiments on Air Quality benchmarks. The experimental

results listed in Table 6 demonstrate that POET consistently surpasses other baselines across all forecasting horizons.

Table 6: MAE results of POET against other position encoding methods on Air Quality benchmarks. The look-back horizon is set to 72H. 3D refers to CARTESIAN3D. We follow the location encoding baselines used (Rußwurm et al., 2024).

| Model | w/o PE | Node | Naive | 3D | RBF | RFF | Theory | Wrap | Sphere | SH | **POET** |
|---|---|---|---|---|---|---|---|---|---|---|---|
| **24H** | 25.933 | 24.131 | 25.108 | 24.197 | 26.093 | 25.065 | 24.856 | 24.164 | 24.651 | 25.083 | **23.194** |
| **48H** | 32.324 | 30.619 | 31.634 | 31.011 | 32.381 | 31.558 | 31.269 | 30.620 | 31.980 | 31.775 | **29.302** |
| **72H** | 36.988 | 35.054 | 36.280 | 35.845 | 37.346 | 36.271 | 36.266 | 35.203 | 36.603 | 36.469 | **33.530** |
| **AVG** | 31.768 | 29.935 | 31.008 | 30.351 | 31.940 | 30.965 | 30.797 | 29.996 | 31.078 | 31.109 | **28.675** |

**Order of Three Attentions Layers**   Additionally, POET adopts Temporal, Spatial, and Variate (TSV) attention layers following the convention in previous works such as TimeSformer (Bertasius et al., 2021), where temporal and spatial attention are applied sequentially in video understanding tasks. This design also aligns with the natural structure of Earth system data, where temporal and spatial relationships are often prioritized before inter-variate dependencies. Therefore, we conducted a comprehensive analysis of the order of these three different layers. As demonstrated in the Table 7, swapping the order of different layers introduces slight variations in the model performance.

Table 7: Ablation study results of POET variants on Air Quality dataset (Beijing) with lookback length $72H$, and forecast length in $\{24H, 48H, 72H\}$. The variate set in the table {Var0, Var1, Var2, Var3, Var4, Var5} corresponds to variable names set {PM2.5, Temperature, Pressure, Humidity, Wind Speed, Wind Direction} respectively.

| Models | | **POET** | POET_VST | POET_STV | POET_SVT | POET_TVS | POET_VTS |
|---|---|---|---|---|---|---|---|
| | 24H | 23.194 | 23.431 | 23.871 | 23.489 | 23.274 | 24.460 |
| Var0 | 48H | 29.302 | 30.011 | 30.141 | 30.075 | 29.161 | 30.306 |
| | 72H | 33.530 | 34.938 | 34.501 | 34.677 | 33.666 | 34.666 |
| | 24H | 3.647 | 3.381 | 3.382 | 3.560 | 3.653 | 3.426 |
| Var1 | 48H | 4.807 | 4.604 | 4.691 | 4.798 | 4.832 | 4.494 |
| | 72H | 4.579 | 4.463 | 4.537 | 4.607 | 4.586 | 4.401 |
| | 24H | 6.688 | 6.317 | 6.605 | 6.626 | 6.559 | 6.561 |
| Var2 | 48H | 9.302 | 9.060 | 9.350 | 9.391 | 9.206 | 9.302 |
| | 72H | 10.095 | 9.895 | 10.094 | 10.202 | 9.955 | 10.261 |
| | 24H | 1.491 | 1.586 | 1.480 | 1.522 | 1.588 | 1.597 |
| Var3 | 48H | 2.303 | 2.375 | 2.293 | 2.334 | 2.384 | 2.401 |
| | 72H | 2.900 | 2.945 | 2.892 | 2.921 | 2.970 | 2.983 |
| | 24H | 5.041 | 4.872 | 5.032 | 5.169 | 5.035 | 4.998 |
| Var4 | 48H | 6.245 | 6.109 | 6.364 | 6.444 | 6.328 | 6.310 |
| | 72H | 6.352 | 6.294 | 6.472 | 6.569 | 6.530 | 6.488 |
| | 24H | 44.948 | 46.303 | 45.000 | 45.072 | 46.119 | 47.821 |
| Var5 | 48H | 55.853 | 56.500 | 56.007 | 55.763 | 56.910 | 58.131 |
| | 72H | 61.326 | 61.213 | 61.057 | 60.789 | 61.907 | 62.983 |
| | 24H | 14.168 | 14.315 | 14.228 | 14.239 | 14.371 | 14.810 |
| | 48H | 17.969 | 18.110 | 18.141 | 18.134 | 18.137 | 18.491 |
| AVG | 72H | 19.797 | 19.958 | 19.925 | 19.961 | 19.936 | 20.297 |
| | Avg | 17.311 | 17.461 | 17.431 | 17.445 | 17.495 | 17.833 |

**Ablation on the latent dimension of the variable position**  In the main text, we introduce a high-dimensional variate position embedding. In this section, we conduct modeling experiments on the positions of 1-dimensional, 2-dimensional, 3-dimensional, and 4-dimensional variables, respectively. The experimental results, presented in Table 8, demonstrate that incorporating high-dimensional coordinates significantly improves model performance compared to lower-dimensional coordinates, highlighting the effectiveness of leveraging high-dimensional positional information.

Table 8: Ablation results on the diverse latent dimension of the variable position.

| Variate Dimension | Flood | | Germany | | Bavaria | | Wind | | Temp | | AVG | |
|---|---|---|---|---|---|---|---|---|---|---|---|---|
| | MSE | MAE | MSE | MAE | MSE | MAE | MSE | MAE | MSE | MAE | MSE | MAE |
| 1D | 0.066 | 0.117 | 0.231 | 0.155 | 0.324 | 0.166 | 3.422 | 0.125 | 5.840 | 0.162 | 1.977 | 0.145 |
| 2D | 0.066 | 0.117 | 0.228 | 0.154 | 0.324 | 0.166 | 3.400 | 0.125 | 5.847 | 0.162 | 1.973 | 0.149 |
| 3D | 0.064 | 0.115 | 0.225 | 0.155 | 0.323 | 0.163 | 3.385 | 0.125 | 5.803 | 0.162 | 1.960 | 0.144 |
| 4D | 0.065 | 0.116 | 0.230 | 0.155 | 0.324 | 0.166 | 3.402 | 0.125 | 5.811 | 0.162 | 1.967 | 0.145 |

**Efficacy of HiPE design in Each Layer**  In our design of POET, we apply three attention layers on temporal-, spatial, variate- level respectively. To validate the effectiveness of each component, we remove each layer respectively. As shown in Figure 7, removing any layer will lead to a decline in the model's forecasting performance, which validates the architectural design of POET.

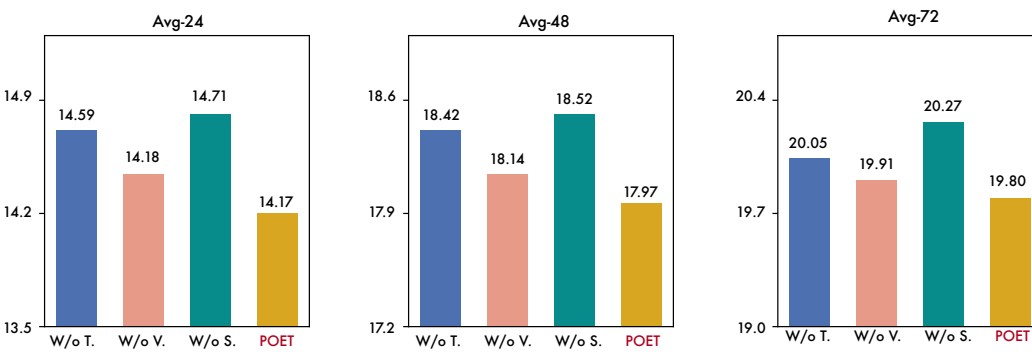

Figure 7: Ablation studies of PORT with various model designs.

# C  MODEL ANALYSIS

**Robustness Analysis**  To validate the robustness of POET under missing data, we applied random masking to the input data by setting certain values to zero, which replicates the challenging real-world scenario with incomplete data and enables a thorough evaluation of forecasting performance. Specifically, we progressively increased the mask ratio to $\{10\%, 15\%, 20\%\}$. The results, summarized in Table 9, demonstrate that POET consistently outperforms other baseline models (PatchTST and iTransformer) across all mask ratios. Moreover, POET exhibits significantly slower performance degradation compared to the baselines, underscoring its robustness and effectiveness in handling missing data.

**More Baselines**  In the main text, we compared POET with state-of-the-art models for time-series forecasting and location encoding. In this section, we add two advanced global weather forecasting models that incorporate station embedding designs as additional baselines (Yang et al., 2024a; Vaughan et al., 2024). As shown in Table 10, POET achieves the best performance on both global and regional benchmarks.

Table 9: Forecasting performance (MAE) of POET, PatchTST, and iTransformer under varying levels of missing data.

| Mast Ratios | | 0% | 10% | 15% | 20% |
|---|---|---|---|---|---|
| POET | 24H | 23.194 | 24.276 | 25.377 | 27.138 |
| | 48H | 29.302 | 30.392 | 31.317 | 32.709 |
| | 72H | 33.530 | 34.952 | 35.817 | 36.959 |
| | AVG | **28.675** | **29.873** | **30.837** | **32.269** |
| PatchTST | 24H | 24.797 | 25.794 | 26.079 | 27.325 |
| | 48H | 31.503 | 31.966 | 32.370 | 33.261 |
| | 72H | 36.091 | 36.640 | 36.904 | 37.694 |
| | AVG | **30.797** | **31.467** | **31.784** | **32.760** |
| iTransformer | 24H | 25.739 | 25.467 | 27.178 | 29.120 |
| | 48H | 32.593 | 31.957 | 33.361 | 34.919 |
| | 72H | 37.077 | 36.649 | 37.928 | 39.289 |
| | AVG | **31.803** | **31.357** | **32.822** | **34.443** |

Table 10: Forecasting Performance of POET against two global weather forecasting baselines.

| Models | | w/oPE | (Yang et al., 2024a) | (Vaughan et al., 2024) | POET |
|---|---|---|---|---|---|
| AirQuality | 24H | 25.993 | 23.611 | 24.741 | **23.194** |
| | 48H | 32.325 | 30.170 | 31.921 | **29.302** |
| | 72H | 36.988 | 34.952 | 35.344 | **33.530** |
| | AVG | 31.768 | 29.577 | 30.336 | **28.675** |
| GTWSF | Wind | 3.878 | 3.550 | 3.599 | **3.385** |
| | Temp | 7.343 | 6.604 | 6.647 | **5.803** |

## D  HYPERPARAMETER SENSITIVITY

In this section, we evaluate the hyperparameter sensitivity of POET on the CausalRivers benchmark's Germany dataset, as illustrated in Figure 8. We vary the number of layers $L \in \{1, 2, 3\}$, the patch size $P \in \{6, 12, 16, 24\}$, and the lookback length $S \in \{48, 72, 96, 192\}$, while strictly fixing the other parameters. We use MAE as the metric for model evaluation. It can be seen that, with the same hyperparameter, the model effect is maintained at almost the same level as the values vary.

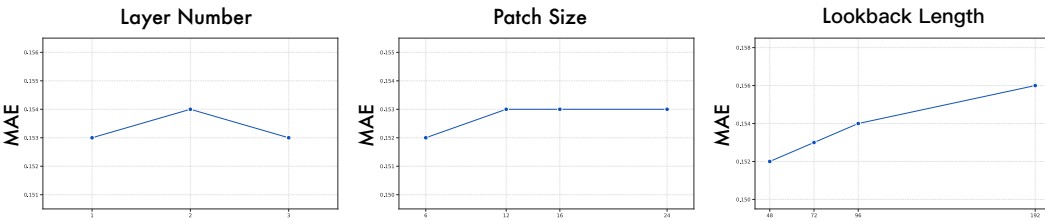

Figure 8: Hyperparameter sensitivity of POET. We use Germany River dataset in the CausalRivers benchmark with a predicted length of 96. The hyperparameters to change are the number of layers $L$, the patch size $P$, and the lookback length $S$.

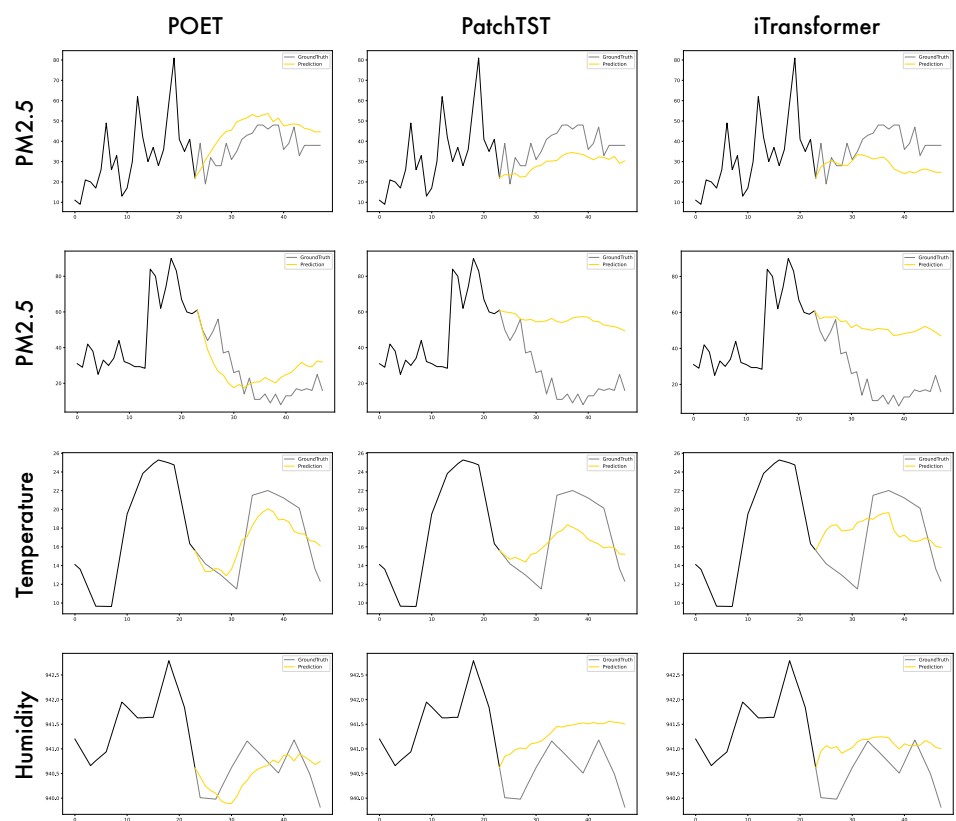

Figure 9: Visualization results on the Air Quality multi-station-multi-variate dataset. We use the prediction experiment setup with input 24 steps and output 24 steps on three variables PM2.5, Temperature, Humidity.

## E    SHOWCASE

For an intuitive comparison across multiple models, we present additional visualizations of the predictions, as shown in Figure 9. On the Air Quality dataset, instead of POET, we randomly selected the showcases from PatchTST and iTransformer, two models that performed well on this dataset. Among the various models, POET predicts results closer to the ground-truth and performs better.

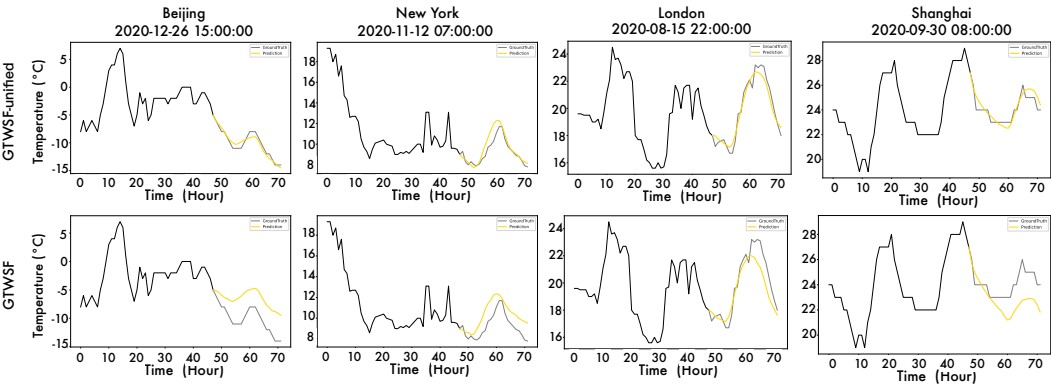

Figure 10: Visualization results for the unified GTWSF dataset and the original separate one.

In addition, we show, separately, the visualization results of POET that predicts the two benchmarks of the GTWSF dataset, as well as the model that jointly predicts the two benchmarks of the same

dataset after combining them together. Due to multivariate considerations, the unified version works better. The visualization is shown in Figure 10.

# F FULL RESULTS

## F.1 FULL RESULTS ON AIR QUALITY

We provide the complete results of the Air Quality dataset, which has multiple stations and variables. We evaluate the predictive performance of POET and other baseline models on this dataset for all variables. Each model is trained using an input of 72 hours (24 steps) and an output of 72 hours (24 steps). Corresponding to the experiment setting in the main text, during the evaluation phase, the forecasting performance is demonstrated by a few different lead times: 24 hours (8 steps), 48 hours (16 steps), and 72 hours (24 steps). We use MAE as an evaluation metric, where a smaller metric indicates better prediction performance. The results are listed in Table 11.

Table 11: Full collaborative forecasting results of Air Quality dataset with lookback length $72H$, and forecast length in $\{24H, 48H, 72H\}$. The variate set in the table {Var0, Var1, Var2, Var3, Var4, Var5} corresponds to variable names set {PM2.5, Temperature, Pressure, Humidity, Wind Speed, Wind Direction} respectively.

| Models | | POET | TimeXer | iTransformer | PatchTST | Crossformer | TiDE | TimesNet | DLinear | SCINet | Autoformer |
|---|---|---|---|---|---|---|---|---|---|---|---|
| Var0 | 24H | **23.194** | 25.463 | 25.739 | 24.797 | 33.962 | 28.490 | 27.802 | 27.542 | 28.553 | 42.888 |
| | 48H | **29.302** | 32.643 | 32.593 | 31.503 | 35.915 | 34.604 | 33.614 | 33.689 | 35.562 | 45.860 |
| | 72H | **33.530** | 37.015 | 37.077 | 36.091 | 37.409 | 38.612 | 39.027 | 37.986 | 40.564 | 47.920 |
| Var1 | 24H | 3.647 | 3.449 | 3.849 | **3.276** | 4.928 | 4.678 | 4.034 | 3.976 | 4.580 | 4.060 |
| | 48H | 4.807 | 4.409 | 4.678 | **4.170** | 5.393 | 5.021 | 4.181 | 4.672 | 4.834 | 4.214 |
| | 72H | 4.579 | 4.333 | 4.445 | **4.120** | 5.432 | 4.728 | 4.192 | 4.501 | 4.917 | 4.375 |
| Var2 | 24H | 6.688 | 7.186 | 7.476 | **6.676** | 8.232 | 8.908 | 9.209 | 8.014 | 7.574 | 9.079 |
| | 48H | 9.302 | 9.678 | 9.520 | 8.785 | 9.520 | 10.129 | 9.954 | 9.625 | **8.763** | 9.814 |
| | 72H | 10.095 | 10.427 | 10.141 | 9.589 | 10.014 | 10.433 | 10.425 | 10.151 | **9.542** | 10.582 |
| Var3 | 24H | **1.491** | 1.837 | 1.805 | 1.721 | 892.874 | 2.138 | 2.316 | 3.198 | 2.549 | 3.373 |
| | 48H | **2.303** | 2.609 | 2.556 | 2.500 | 893.064 | 2.786 | 2.819 | 4.506 | 3.116 | 3.675 |
| | 72H | **2.900** | 3.167 | 3.099 | 3.062 | 892.836 | 3.258 | 3.385 | 5.391 | 3.601 | 4.083 |
| Var4 | 24H | 5.041 | 4.974 | 5.150 | **4.739** | 5.620 | 5.789 | 6.424 | 5.235 | 7.107 | 6.388 |
| | 48H | 6.245 | 6.154 | 6.110 | **5.839** | 6.112 | 6.407 | 6.852 | 6.060 | 7.621 | 6.556 |
| | 72H | 6.352 | 6.308 | 6.162 | **5.992** | 6.226 | 6.360 | 6.870 | 6.121 | 7.693 | 6.605 |
| Var5 | 24H | **44.948** | 46.068 | 48.524 | 45.820 | 132.951 | 55.996 | 55.945 | 50.549 | 56.733 | 62.678 |
| | 48H | **55.853** | 56.858 | 59.201 | 57.465 | 132.599 | 63.214 | 63.354 | 59.105 | 65.805 | 67.462 |
| | 72H | **61.326** | 62.231 | 63.693 | 62.294 | 132.364 | 66.389 | 66.585 | 62.871 | 69.315 | 70.263 |
| AVG | 24H | **14.168** | 14.829 | 15.424 | 14.505 | 179.761 | 17.667 | 17.621 | 16.419 | 17.849 | 21.411 |
| | 48H | **17.969** | 18.725 | 19.110 | 18.377 | 180.434 | 20.360 | 20.129 | 19.610 | 20.950 | 22.930 |
| | 72H | **19.797** | 20.580 | 20.769 | 20.191 | 180.713 | 21.630 | 21.747 | 21.170 | 22.605 | 23.971 |
| | Avg | **17.311** | 18.045 | 18.434 | 17.691 | 180.303 | 19.886 | 19.832 | 19.066 | 20.468 | 22.771 |
| 1st Count | | **12** | 0 | 0 | 7 | 0 | 0 | 0 | 0 | 2 | 0 |

## F.2 FULL RESULTS ON CAUSALRIVERS BENCHMARKS

As a supplement to the main text, we provide the full results of CausalRivers in Table 12.

Table 12: Full results of the forecasting task on the CausalRivers benchmark. A lower MSE or MAE indicates a better prediction. We follow the standard protocol of long-term forecasting (Wang et al., 2024b), where both the input length and prediction length are set to 96 for all baselines. Avg means the average results from all three datasets: Flood, Germany and Bavaria. "-" denotes the out-of-memory (OOM) problem.

| Model | POET | | TimeXer | | iTransformer | | PatchTST | | Crossformer | | TiDE | | TimesNet | | DLinear | | SCINet | | Autoformer | |
|---|---|---|---|---|---|---|---|---|---|---|---|---|---|---|---|---|---|---|---|---|
| Metric | MSE | MAE | MSE | MAE | MSE | MAE | MSE | MAE | MSE | MAE | MSE | MAE | MSE | MAE | MSE | MAE | MSE | MAE | MSE | MAE |
| Flood | **0.064** | **0.115** | 0.066 | 0.116 | 0.070 | 0.123 | 0.068 | 0.117 | 0.095 | 0.181 | 0.077 | 0.138 | 0.081 | 0.145 | 0.096 | 0.212 | 0.078 | 0.139 | 0.161 | 0.320 |
| Germany | **0.225** | 0.155 | 0.236 | 0.154 | 0.235 | 0.154 | 0.229 | **0.153** | 0.240 | 0.184 | 0.242 | 0.159 | 0.294 | 0.191 | 0.235 | 0.188 | 0.312 | 0.192 | 0.381 | 0.311 |
| Bavaria | **0.323** | **0.163** | 0.335 | 0.165 | 0.344 | 0.166 | 0.345 | 0.169 | 0.344 | 0.202 | - | - | - | - | 0.383 | 0.211 | 0.434 | 0.187 | 1.660 | 0.840 |
| AVG | **0.204** | **0.144** | 0.212 | 0.145 | 0.217 | 0.148 | 0.214 | 0.147 | 0.226 | 0.189 | - | - | - | - | 0.238 | 0.204 | 0.275 | 0.173 | 0.734 | 0.490 |

