# OpenReview forum: "POET: Partially Observed Earth Transformer with High-Dimensional Position Embedding"
_ICLR.cc/2026/Conference — Submitted to ICLR 2026_

### Official Review · Reviewer_AXCp · 2025-10-23

**Soundness:** 3
**Presentation:** 3
**Contribution:** 2
**Rating:** 4
**Confidence:** 4

**Summary:**

SUMMARY: The authors tackle the problem of encoding spatio-temporal Earth system data (think climate, atmospheric data etc.), focusing specifically on the task of dense reconstruction from sparse / partial observations. For example, this could be interpolating (and potentially extrapolating in time) a dense spatio-temporal map of the world from observations at a set of sensor stations. This is a well established methodological problem and relevant to a myriad of applications in the geosciences and beyond. The methodological contribution of the paper is twofold: (1) the authors propose a new positional encoding method HiPE to encode complex spatio-temporal interdependencies; (2) on top of that, they propose POET, a learning framework with an attention mechanism, for modeling spatio-temporal data. They test their proposed method on real-world Earth systems data including weather and air quality data.

**Strengths:**

STRENGTH:

- The motivation for the paper and the problem it tackles is extremely relevant, has many concrete applications and direct pathways for impacting real-world modeling and decision making. It's always great to see more work on geospatial data at the major ML conferences!

- Building better neural net architectures for spatio-temporal data is a very important problem and e.g. explicitly mentiond in [1]; this is especially true for neural nets that take geographic coordinates and/or timestamps as inputs. The proposed neural net architecture is explained well.

- I enjoyed the ablation studies in the paper, especially the analysis of cross-correlations between different "variates" / input channels.

- Overall, the paper is written quite well and somewhat easy to follow (except the details on the positional encoding)

**Weaknesses:**

SHORTCOMINGS:

Major:

- My main problem with this study is the following: In their discussion of related work, the authors state that "To date, the encoding of positional information in the context of complex Earth system modeling remains an underexplored
area, largely due to the high dimensionality and unclear spatiotemporal dependency." This statement is not correct in its  gravity and the related work section lacks discussion of several existing positional embedding approaches for Earth data. For example, work on positional encodings for geographic coordinates (lon, lat) includes work on sinusoidal transforms ([2,3]) and spherical harmonics transforms ([4]). Work on jointly encoding spatio-temporal coordinates includes [5]. And work on positional encodings for different "variates" (i.e., channels or spectral bands) includes [6, 7]. The lack of engagement with any of this work and the subsequent lack of testing and comparing against these different approaches makes it difficult for me to assess how useful the HiPE contribution actually is.

- Following from this, I am not fully sure how the spatio-temporal coordinates are actually encoded. If I understand Eq 3 correctly, they are just left raw? That is, raw lon/lat coordinates and a raw timestamp, with 2 and 1 learnable parameters respectively? If that is indeed the case, the learning of spatio-temporal dynamics might be severely limited. Or am I missing something here? I would appreciate a clarification.

Minor:

- The results tables are partially quite hard to read, especially Tab 2 would benefit from reformatting.

**Questions:**

Overall, this paper tackles an important problem and is presented well. What holds this back is the lack of discussion and comparison to previous work on space-time coordinate encoding. I would appreciate some clarifications from the authors especially on the points raised in "Weaknesses" above.

References for papers mentioned in the above boxes:

[1] Rolf, Esther, et al. "Mission Critical--Satellite Data is a Distinct Modality in Machine Learning." arXiv preprint arXiv:2402.01444 (2024).

[2] Mai, Gengchen, et al. "Sphere2Vec: multi-scale representation learning over a spherical surface for geospatial predictions." arXiv preprint arXiv:2201.10489 (2022).

[3] Mai, Gengchen, et al. "Multi-scale representation learning for spatial feature distributions using grid cells." arXiv preprint arXiv:2003.00824 (2020).

[4] Rußwurm, Marc, et al. "Geographic location encoding with spherical harmonics and sinusoidal representation networks." arXiv preprint arXiv:2310.06743 (2023).

[5] Chen, Weibin, et al. "Deep random features for scalable interpolation of spatiotemporal data." The Thirteenth International Conference on Learning Representations. 2024.

[6] Cong, Yezhen, et al. "Satmae: Pre-training transformers for temporal and multi-spectral satellite imagery." Advances in Neural Information Processing Systems 35 (2022): 197-211.

[7] Ahmad, Muhammad, et al. "Spatial spectral transformer with conditional position encoding for hyperspectral image classification." IEEE Geoscience and Remote Sensing Letters (2024).

---

> ### Author Response · Authors · 2025-11-24
> **Response to Reviewer AXCp (Part 1)**
>
> Many thanks to Reviewer AXCp for providing a detailed and in-depth review, which helped us significantly improve the quality of our submission.
>
>
> > **W1:** This related work lacks a discussion of several existing positional embedding approaches for Earth data. For example, work on positional encodings for geographic coordinates (lon, lat) includes work on sinusoidal transforms ([2,3]) and spherical harmonics transforms ([4]). Work on jointly encoding spatio-temporal coordinates includes [5]. And work on positional encodings for different "variates" (i.e., channels or spectral bands) includes [6, 7]. Lack of engagement with any of this work and the subsequent lack of testing and comparing against these different approaches.
>
> Thank you for the valuable suggestions. We have included them in $\underline{\text{Section 2 of the revised paper}}$ and expanded our experiments to include these positional encoding approaches as baselines. Technologically, we retain the Transformer design of POET but replace the RoPE mechanism in the attention layers with the alternative location encoding modules. We conducted comprehensive ablation studies on both the meteorological forecasting and air quality forecasting tasks. The results are summarized in the tables below. It can be observed that location encoding methods generally outperform the model without positional encoding (w/oPE), which underscores the value of integrating spatial information into Earth system modeling. Notably, the proposed POET consistently outperforms the alternatives. We have included the experimental results in $\underline{\text{Table 4 and Table 6 of the revision}}$.
>
> | Meteorological | w/oPE | Direct | 3D | RBF | RFF | Theory | Wrap | Sphere | SH | POET |
> |:---:|:---:|:---:|:---:|:---:|:---:|:---:|:---:|:---:|:---:|:---:|
> | Wind | 3.878 | 3.893 | 3.895 | 3.868 | 3.520 | 3.854 | 3.896 | 3.515 | 3.938 | 3.385 |
> | Temp | 7.343 | 7.574 | 7.762 | 7.289 | 6.132 | 7.928 | 7.917 | 6.499 | 8.376 | 5.803 |
>
>
> | Air Quality | w/oPE | Direct | 3D | RBF | RFF | Theory | Wrap | Sphere | SH | POET |
> |:---:|:---:|:---:|:---:|:---:|:---:|:---:|:---:|:---:|:---:|:---:|
> |24H | 24.000 | 25.993 | 24.197 | 31.011 | 26.093 | 25.065 | 24.856 | 24.164 | 24.651 | 25.083 |
> |48H| 32.324 | 35.657 | 31.011 | 32.381 | 31.558 | 31.269 | 30.620 | 31.980 | 31.775 | 29.302 |
> |72H | 36.988 | 33.240 | 35.845 | 37.3461 | 36.271 | 36.266 | 35.203 | 36.603 | 36.469 | 33.530 |
> |AVG| **31.768** | **33.240** | **30.351** | **31.940** | **30.965** | **30.797** | **29.996** | **31.078** | **31.109** | **28.675** |

---

> ### Author Response · Authors · 2025-11-24
> **Response to Reviewer AXCp (Part 2)**
>
> > **W2:** Following from this, I am not fully sure how the spatio-temporal coordinates are actually encoded. If I understand Eq 3 correctly, they are just left raw? That is, raw lon/lat coordinates and a raw timestamp, with 2 and 1 learnable parameters respectively? If that is indeed the case, the learning of spatio-temporal dynamics might be severely limited. Or am I missing something here? I would appreciate a clarification.
>
> Thank you for your insightful question. In POET, we follow prior works [1] by leveraging the attention mechanism to model spatial relationships. Instead of designing a complex location encoding module, we use Rotary Position Embedding (RoPE), which encodes absolute positions into the self-attention mechanism and enables it to decay inter-token dependencies with increasing relative distances. Concretely, for a token at position m, RoPE divides the token's d-dimensional hidden space into d/2 sub-spaces and rotates each 2‑dimensional feature subspace based on multi‑frequency sinusoidal phases. For the $i$‑th frequency, the rotation angle is $\theta_i = 10000^{-2(i-1)/d}$, where $i \in [1, 2, ..., d/2]$ and the corresponding rotation matrix is
> $R(m)\_i = [[cos(m\theta_i), −sin(m\theta_i)],
> 		   [sin(m\theta_i), cos(m\theta_i)]].$
>
> A key property of RoPE is that $R(m)^T R(n) = R(n-m)$, meaning that the **attention between tokens at positions m and n is governed by their relative offset rather than absolute indices**. With multi‑frequency rotations, RoPE encodes relative positional relationships across multiple spatial scales. The resulting attention score is calculated as follows:
> $q_m^Tk_n = (R(m)W_qx_m)^T (R(n)W_kx_n) = x_m^TW_qR(n-m)W_kx_n$, which makes the model inherently sensitive to relative distances.
>
> In our proposed HiPE, as we stated in $\underline{\text{Equation (4) of the original paper}}$, we divide the hidden representation into $C$ subspaces, where $C$ equals the dimensionality of the positional information, and independently apply RoPE on each subspace with the corresponding dimension of the position information.
>
> Therefore, the raw coordinates are not a limitation; they **provide valuable positional priors that guide the attention mechanism in learning spatial correlations**. Furthermore, the learnable positional components are data-driven, allowing the model to uncover hidden relationships within the observed data that may not be explicitly described by existing knowledge.
>
> [1] Bertasius G, Wang H, Torresani L. Is space-time attention all you need for video understanding?. ICML 2021.
>
> > **W3:** The results tables are partially quite hard to read, especially Tab 2 would benefit from reformatting.
>
> Thank you for the valuable suggestion. We have revised the format of tables in our resubmission.

---

### Official Review · Reviewer_pGZZ · 2025-10-27

**Soundness:** 3
**Presentation:** 4
**Contribution:** 2
**Rating:** 6
**Confidence:** 4

**Summary:**

The authors propose a general mulit-variate spatio-temporal machine learning model called POET to forecast Earth Systems.
The model proposes to express earth models as a three dimensional setup, where one dimension is the spatial position (lat, lon), the second one time, and the third one the variate of interest.
The model is comprised of three sequential transformer blocks, with each block receiving the positional encoding of one of the three dimension, effectively attending to each dimension separately.
These layers are called "Temporal-Attention", "Spatial-Attention" and "Variate-Attention".
At each layer, they additionally use a extended version of RoPE that allow for a better Rotary position embedding, which translates the absolute positional embeddings into a relative one, and allows the model to learn from the relation between grid positions.
Tested on three earth system tasks, the model outperforms a series of of models and shows versatility and adaptability.

**Strengths:**

The article is well written and the figures of high quality, it is easy to follow. It is easy to see how the proposed model could fit a wide range of earth system applications. The model can generalize well to a wide range of applications and is showcased on earth system tasks that are quite different from each other.
Integrating the positional encoding in a layers is smart, and I'm surprised this has not been done before. Intuitively, it makes sense to apply RoPE in this context. In addition, utilizing RoPE instead of a relative positional encoding seems quite promising.

**Weaknesses:**

Although the paper is interesting, there are, in my opinion, a few key weaknesses:
- It is not clear how the data was split for the downstream tasks. The model obviously has to be trained from scratch for each downstream task, so there should be a train, validation and test dataset. Given that the model is a forecasting task, there should be a holdout set that has never been seen by the model. From the result, it is not clear if that is the case, and we can't make sure there is no data leakage.
- The model is clearly a fixed location forecast with spatio-temporal dynamics. But the authors only compare to state of the art models of time series forecasting, not spatio-temporal forecasts. I do think this is a weakness as spatio-temporal models for fixed location forecasts have consistently shown better performance than time series forecasts for applications like weather forecasting (c.f. Yang et al. 2025 (https://arxiv.org/abs/2410.12938), Allen et al. 2025 (https://arxiv.org/abs/2404.00411)). Integrating spatio-temporal models as baseline comparison would be welcome.
- It is not clear what the authors mean by "Knowledge driven position". The authors mention "leverages prior information about the observation. This prior knowledge is derived from domain expertise, structured metadata, or predefined rules, which provide static and interpretable position information", but there is no mention in the remaining paper of the use metadata, rules etc. This is quite confusing to me.
- Although I do think that the proposed positional encoding is very interesting, it isn't shown in the paper that this is actually performing better than the alternatives. Does the model perform worse if all positional encodings were introduce at the same time? Is RoPE performing better than a learned positional encoding or a relative positional encoding?


Minor weakness:
- Although the figures are of very high quality, they are a bit misleading. The use of the globe in figures 1 and 2 make it seem like the model is either 1) global or 2) allows inference at arbitrary locations. But after reading the paper it is clear that the model is a fixed location forecasting model. I am not sure how, but I would try to make this more clear.
- Why only show MAE for certain results but both RMSE and MAE for others?
- In table 2, Germany's MAE is better for TimeXer, iTransformer, PatchTST. Wrongly bolded for POET.
- Please keep the formatting consistent between tables

**Questions:**

What is the prior knowledge?
What is the data split for use cases?
Can you compare to performance if positional encoding was done all at once?
What about using a learned positional encoding, or a relative instead of absolute?

---

> ### Author Response · Authors · 2025-11-24
> **Response to Reviewer pGZZ (Part 1)**
>
> Many thanks to Reviewer pGZZ for providing a detailed review and recognizing our contributions.
>
> > **W1 & Q2:**  It is not clear how the data was split for the downstream tasks. There should be a train, validation and test dataset. Given that the model is a forecasting task, there should be a holdout set that has never been seen by the model. From the result, it is not clear if that is the case, and we can't make sure there is no data leakage.
>
> Sorry for the missing details. We follow the standard protocol that divides each dataset into the training, validation, and testing subsets according to the chronological order. Specifically, **the train/val/test ratio is 7:1:2 for all datasets, which are aligned to their original paper [1,2]**. Thus, there is no data leakage in our experiments.
>
> [1] Liu, Yong, et al. "Timer-xl: Long-context transformers for unified time series forecasting." ICLR 2025.
>
> [2] Hettige, Kethmi Hirushini, et al. "Airphynet: Harnessing physics-guided neural networks for air quality prediction." ICLR 2024.
>
> > **W2:** The authors only compare to state-of-the-art models of time series forecasting, not spatio-temporal forecasts. I do think this is a weakness as spatio-temporal models for fixed location forecasts have consistently shown better performance than time series forecasts for applications like weather forecasting (c.f. Yang et al. 2025, Allen et al. 2025). Integrating spatio-temporal models as baseline comparison would be welcome.
>
> Thanks for your valuable suggestion. We would like to highlight that in Air Quality forecasting experiments, we **have already included seven deep spatiotemporal models as our baselines** in $\underline{\text{Table 1 of the original paper}}$. Following your suggestion, we include these two approaches as additional baselines, and the results are listed below.
>
> |Method|w/oPE |Yang et al. 2025|Allen et al. 2025| POET|
> |:-------: |:-------: |:-------: |:-------: |:-------: |
> | 24H   |25.993|23.611   | 24.741  |23.194|
> | 48H   |32.325|30.170   | 31.921  |29.302|
> | 72H   |36.988|34.952   | 35.344  |33.530|
> | AVG  |**31.768**|**29.577**|**30.336**|**28.675**|
>
> > **W3 & Q1:** It is not clear what the authors mean by "Knowledge driven position". The authors mention "leverages prior information about the observation. This prior knowledge is derived from domain expertise, structured metadata, or predefined rules, which provide static and interpretable position information", but there is no mention in the remaining paper of the use metadata, rules etc.
>
> Sorry for the missing clarification. The "knowledge-driven position" refers to static, pre-existing information about each observation, which is external to the time series data itself and provides valuable insights into the prediction. In the context of our experiments, this information corresponds to the geographical coordinates (latitude and longitude) of each station, which is provided as metadata in the Earth Observation datasets we used in Section 4.
>
> To enhance clarity, we have revised the term "knowledge-driven position" to "prior position" in the revised submission and have provided more description about the prior position in $\underline{\text{Line 217 and 309 of the revised  paper}}$.
>
> > **W4 & Q3 & Q4:** Although I do think that the proposed positional encoding is very interesting, it isn't shown in the paper that this is actually performing better than the alternatives. Does the model perform worse if all positional encodings were introduced at the same time? Is RoPE performing better than a learned positional encoding or a relative positional encoding?
>
> Thanks for your insightful suggestion. We have newly conducted an ablation study with a different position encoding strategy. The results are listed below, and the learned position encoding method follows [1]. POET consistently outperforms the alternatives in average performance, achieving the best results across all horizons. Introducing all positional encodings simultaneously (POET + Absolute PE) slightly degrades performance, likely due to redundant or conflicting spatial information.
>
> |Method|POET      |Absolute PE|Relative PE| Learned PE|POET+Absolute PE|
> |:-------: |:-------: |:-------: |:-------: |:-------: | :-------: |
> | 24H   |23.194    | 24.102   | 23.584  | 24.131 | 23.514 |
> | 48H   |29.302    | 30.896   | 30.651  | 30.619 | 29.489 |
> | 72H   |33.530    | 35.798   | 35.716  | 30.054 | 33.701 |
> | AVG  |**28.675**|**30.265**|**29.984**|**29.935**|**28.901**|
>
> [1] Taming Local Effects in Graph-based Spatiotemporal Forecasting.

---

> ### Author Response · Authors · 2025-11-24
> **Response to Reviewer pGZZ (Part 2)**
>
> > **W5:** Although the figures are of very high quality, they are a bit misleading. The use of the globe in figures 1 and 2 make it seem like the model is either 1) global or 2) allows inference at arbitrary locations. But after reading the paper it is clear that the model is a fixed location forecasting model. I am not sure how, but I would try to make this more clear.
>
> Thank you for pointing out the potential issues. Figure 1 illustrates the high dimensionality of partially observed Earth data, and Figure 2 shows how POET integrates positional information into the attention mechanism to capture intricate dependencies in this high‑dimensional space.
>
> In the real world, stations are located at fixed geographic positions. Accordingly, POET is a fixed-location forecasting approach that encodes positional priors for observations collected at a set of fixed sensors/stations. These stations can be distributed globally (Section 4.1) or locally (Sections 4.2 and 4.3), but POET itself does not perform continuous spatial interpolation or direct inference at arbitrary coordinates. We agree that enabling inference at arbitrary locations is an interesting and valuable research direction, and we will explore it in future work.
>
>
> > **W6:** Why only show MAE for certain results but both RMSE and MAE for others?
>
> Sorry for the missing explanation. For each dataset, we adhere to its canonical evaluation metric as reported in prior studies.
>
> > **W7 & W8:** In table 2, Germany's MAE is better for TimeXer, iTransformer, PatchTST. Wrongly bolded for POET. Please keep the formatting consistent between tables.
>
> Thank you for highlighting the potential issue in Table 2.  We have revised the format of tables in our resubmission, as well as this typo.

---

### Official Review · Reviewer_UAmW · 2025-10-27

**Soundness:** 3
**Presentation:** 3
**Contribution:** 2
**Rating:** 4
**Confidence:** 4

**Summary:**

In this paper, the authors present POET, for Partially Observed Earth Transformer, an architecture for spatio-temporal Earth modelling when available data is not evenly distributed across the globe and consistently available across sensors. The architecture consists of an encoder with three self attention layers to capture  temporal, spatial, and variate dependencies. The authors introduce a new high dimensional positional encoding strategy that encodes the geographical bias of Earth positions.
The positional encoding consists of an absolute position (rules-based/expert derived) and a learnable position. The authors demonstrate their method on meteorological, air quality  and river discharge forecasting.

**Strengths:**

The paper is clear and well-written. The paper tackles a challenge  that is common in many Earth modelling tasks, the fact that data is not available on a regular grid on the Earth and that there are spatial biases.
The experiments are sound and the authors compare their method thoroughly to other baselines on 3 different tasks. The model demonstrates improved performance on all 3 datasets, and the authors also conduct thorough ablation studies on the components of their model.

**Weaknesses:**

I think the name "Partially-observed earth transformer" is a bit misleading. For example, it is very common for certain meteorological data to only be available at meteorological stations, which obviously are not evenly distributed across the globe.
With respect to encoding space and time, the authors do not mention any work in the location embedding space (cf. geopriors, location embeddings such as SatCLIP, deep random features...).
I think the main thing is: what is the problem that POET tackles that work in location embeddings is not really considering?
"Partially observed" might be more relevant if it referred to missing data points at certain times/locations (which seemed to be the case that was considered when looking at Fig.1. However when looking at the design of the model and the evaluation, it seems that all variate values are available at a given time and location.

**Questions:**

Related to the point raised in the weaknesses, how do you deal with missing data of certain variables at certain time/lovations? Have the authors tried experiments where information is not available at all times across all sites of a considered dataset? If not, can you explain how your model could be used in that context of if not, state explicitly this limitation?

Have you tried space time encoding, such as the ones used in torchspatial ?
Have you tried swapping T,S,V order in the encoder?

There seems to be a discrepancy in POET AVG results between tables 2 and 3. Which ones are the results to consider?
Table 3 results: do the authors have some idea why the use of HiPE (or partial components of HiPE) do not bring a big improvement compared to not using it on the Flood, Germany and Bavaria CausalRIvers predictions, when HiPE seems more useful for the Wind and Temp predictions of the GTWSF dataset?

Analysis of spatial proximity: how are the spatial embeddings reprojected into lat lon on the map? I don’t really understand what the point of mapping them back on the map is? Is that more to compare the distance between the expert-derived embeddings vs distance between the hiPE embeddings, and in that case why not represent them in the embedding space?

Relatedly, I have a comment about Fig. 5 If I understand correctly, the results that are compared are from the POET model with HiPE and without HiPE. Regarding this claim, “Therefore, the relative distance between the spatial positions learned by POET is significantly greater than their prior geographical position” I’m not sure it is very visible from the map that the distance is significantly greater between the red triangles than the blue dots.

I may have missed this but I could not find how the knowledge-driven position is defined in the different experiments.

Overall, I think the paper has potential, the experiments are sound and there was clearly a lot of work put into this. But so far it is not quite clear how well motivated proposing this architecture is, and there are some missing comparisons with simple space time encoding schemes. There are also some missing details about the implementation (what is the knowledge-driven expertise). I am very open to revising my score, I am sorry if I missed something here, and I would appreciate if the authors would clarify these points.

---

> ### Author Response · Authors · 2025-11-24
> **Response to Reviewer UAmW (Part 1)**
>
> Many thanks to Reviewer UAmW for providing a detailed review and recognizing our contributions.
>
> > **W1.1 & Q2.1:**  The authors do not mention any work in the location embedding space. Have you tried space-time encoding, such as the ones used in torchspatial?
>
> We have newly included location encoding into $\underline{\text{Section 2 of the revised paper}}$.
>
> As per your request, we have **newly included several location encoding baselines** based on torchspatial and Sphere2Vec [1,2], namely  Naive, RBF, RFF, and Sphere. Technologically, we maintain the Transformer design of POET while replacing the RoPE mechanism in the attention layers with different location encoding modules. The results, as shown below, demonstrate that the inclusion of position information enhances forecasting performance, with POET consistently achieving the best performance across all forecasting horizons. We have included the experimental results in $\underline{\text{Table 4 of the revision}}$.
>
> |Method|w/oPE |Naive|Sphere|RBF|RFF| POET|
> |:-------: |:-------: |:-------: |:-------: |:-------: |:-------: |:-------: |
> | 24H   |25.993|25.108   | 24.651  | 26.093 | 25.065 |23.194|
> | 48H   |32.325|31.634   | 31.980  | 32.381 | 31.558 |29.302|
> | 72H   |36.988|36.280   | 36.603  | 37.346 | 36.271 |33.530|
> | AVG  |**31.768**|**31.008**|**31.078**|**31.940**|**30.965**|**28.675**|
>
> [1] Wu, Nemin, et al. "Torchspatial: A location encoding framework and benchmark for spatial representation learning." NeurIPS 2024.
>
> [2] Mai, Gengchen, et al. "Sphere2Vec: multi-scale representation learning over a spherical surface for geospatial predictions." arXiv preprint arXiv:2201.10489 (2022).
>
> > **W1.2:** The name "Partially-observed earth transformer" is a bit misleading. What is the problem that POET tackles that work in location embeddings is not really considering?
>
> Sorry for the missing explanation. The term "Partially-observed" follows the description in [1], which highlights that "the scattered stations only provide partial observations governed by the continuous space–time global weather system, thus introducing thorny challenges to worldwide forecasting." Therefore, these observations are typically recorded as high-dimensional time series, where the positional information is multifaceted, spanning spatial, variate, and temporal dimensions. Previous work on location embeddings primarily focuses on encoding the geographical location of image data, and misses the complex cross-dimensional dependencies of the Earth time series data.
>
> In the proposed HiPE, we address these challenges by unifying the positional information across dimensions into two complementary components: prior position and learnable position. This unified representation is then seamlessly incorporated into the attention layers via Rotary Position Embedding, enabling POET to capture intricate dependencies within the partially observed Earth system effectively.
>
> [1] Wu, Haixu, et al. "Interpretable weather forecasting for worldwide stations with a unified deep model." Nature Machine Intelligence 5.6 (2023): 602-611.

---

> ### Author Response · Authors · 2025-11-24
> **Response to Reviewer UAmW (Part 2)**
>
> > **Q1:** How do you deal with missing data of certain variables at certain time/lovations? Have the authors tried experiments where information is not available at all times across all sites of a considered dataset? If not, can you explain how your model could be used in that context of if not, state explicitly this limitation?
>
> Thank you for raising this insightful question. Missing data is indeed a common challenge in real-world Earth system observations, often resulting in incomplete or zero-valued entries in the data. As per your request, we applied random masking to the input data by setting certain values to zero, which replicates this challenging scenario and allows us to evaluate forecasting performance. Specifically, we progressively increased the mask ratio to 10\%, 15\%, 20\%, respectively. The results are listed as follows, where POET consistently outperforms other baselines (PatchTST and iTransformer) across all mask ratios, exhibiting significantly slower performance degradation compared to the others. These findings highlight POET's robustness and effectiveness in handling missing data, making it well-suited for real-world applications.
>
> | Mask Ratio  |      | 0%      | 10%    | 15%    | 20%    |
> |:--------------: |:----:|:-------:|:--------:|:-------:|:-------:|
> | **POET**        | 24H   | 23.194  | 24.276 | 25.377   | 27.138  |
> |                 | 48H   | 29.302  | 30.392 | 31.317   | 32.709  |
> |                 | 72H   | 33.530  | 34.952 | 35.817   | 36.959  |
> |                 | AVG  |**28.675**|**29.873**|**30.837**|**32.269**|
> | **PatchTST**    | 24H   | 24.797  | 25.794 |26.079   | 27.325  |
> |                 | 48H   | 31.503  | 31.966 |32.370   | 33.261  |
> |                 | 72H   | 36.091  | 36.640 |36.904   | 37.694  |
> |                 | AVG  |**30.797**|**31.467**|**31.784**|**32.760**|
> | **iTransformer**| 24H   | 25.739  | 25.467 |27.178   | 29.120  |
> |                 | 48H   | 32.593  | 31.957 |33.361   | 34.919  |
> |                 | 72H   | 37.077  | 36.649 |37.928   | 39.289  |
> |                 | AVG  |**31.803**|**31.357**|**32.822**|**34.443**|
>
> > **Q2.2:** Have you tried swapping T,S,V order in the encoder?
>
> We have already included an ablation of the order of three attention layers in $\underline{\text{Table 6 in the Appendix of the original submission}}$. The experimental results indicate that swapping the order of different layers introduces slight variations in model performance.
>
> > **Q3.1:** There seems to be a discrepancy in POET AVG results between tables 2 and 3. Which ones are the results to consider?
>
> Sorry for the missing clarification. We would like to highlight that Table 2 reports the forecasting performance specifically on the CausalRivers benchmark, which includes the Flood, Germany, and Bavaria datasets. In contrast, Table 3 presents an ablation study across a broader set of datasets, combining CausalRivers and the Meteorological benchmarks in Figure 3. As a result, the AVG in Table 3 represents the average performance across all five datasets, and **the specific results for POET reported in these two tables are consistent within the respective datasets**.
>
> > **Q3.2:** Do the authors have some idea why the use of HiPE (or partial components of HiPE) do not bring a big improvement compared to not using it on the Flood, Germany and Bavaria CausalRIvers predictions, when HiPE seems more useful for the Wind and Temp predictions of the GTWSF dataset?
>
> Thank you for this insightful question regarding the differential impact of HiPE across benchmarks. The smaller gains from HiPE on the Flood, Germany, and Bavaria datasets can be attributed to the inherent geographical concentration of these regional datasets as visualized in $\underline{\text{Figure 6 of the original submission}}$.
>
> In contrast, global datasets like Wind and Temp in GTWSF have globally distributed stations with significant location discrepancies. In such cases, the inclusion of HiPE enables the model to focus more on stations that are geographically closer, resulting in substantial performance improvements. This distinction further underscores the versatility and scalability of HiPE, particularly in handling datasets with greater spatial diversity.

---

> ### Author Response · Authors · 2025-11-24
> **Response to Reviewer UAmW (Part 3)**
>
> > **Q4:** Analysis of spatial proximity: how are the spatial embeddings reprojected into lat lon on the map? I don’t really understand what the point of mapping them back on the map is? Is that more to compare the distance between the expert-derived embeddings vs distance between the hiPE embeddings, and in that case why not represent them in the embedding space?
>
> As outlined in $\underline{\text{Equation 3 of the original submission}}$, the learnable position is added directly to the prior position and embedded via RoPE directly into the self-attention mechanism, rather than embedded into a "position embedding space."
>
> Mapping the learned positions back onto the map serves an interpretability purpose, as it **reveals the hidden relationships within the observed data that may not be explicitly described by the prior positions**. Based on these considerations, we provide a visualization analysis in $\underline{\text{Figure 4 (Right)}}$, which shows the positional shifts of stations situated along the eastern coast of the United States. These learned position shifts are influenced by the scarcity of stations in the ocean, reflecting data-driven spatial information that is not inherent in the prior geographic positions.
>
> > **Q5:** It is not very visible in Figure 5 from the map that the distance is significantly greater between the red triangles than the blue dots.
>
> Thanks for your valuable observations. We would like to clarify that the relative visibility of the positional shifts in Figure 5 is indeed influenced by the spatial scale of the map. Specifically, the Beijing and Dalian stations are originally separated by a significant distance of approximately **466.5 kilometers**, which causes the individual positional shifts after integrating the learnable positions to appear less pronounced on the map.
> - For the Dalian station, its coordinates shifted from(38.9657∘N,121.5387∘E)to(39.1933∘N,121.7693∘E), resulting in a movement of approximately **28.2 kilometers**.
>
> - For the Beijing station, its coordinates shifted from(40.0801∘N,116.5847∘E)to(39.9618∘N,116.2708∘E), resulting in a movement of approximately **30.6 kilometers**.
>
> The visualization in Figure 5 highlights that the shifts effectively improve the forecasting performance of the model.
>
> > **Q6:** I may have missed this but I could not find how the knowledge-driven position is defined in the different experiments.
>
> Sorry for the missing explanation. The knowledge-driven position refers to the prior information about observation, which typically appears as metadata in the dataset. In the context of our experiments, it corresponds to the chronological information in the temporal dimension and the geographical location of each station in the spatial dimension. To enhance clarity, we have revised the term "knowledge-driven position" to "prior position" in $\underline{\text{Line 217 of the revised paper}}$ and provided more description on the prior position used in experiments in $\underline{\text{Line 309}}$.

---

### Official Review · Reviewer_2Zqh · 2025-10-29

**Soundness:** 3
**Presentation:** 3
**Contribution:** 3
**Rating:** 6
**Confidence:** 5

**Summary:**

The authors propose POET, a Transformer-based model that alternately captures temporal, spatial, and cross-variable dependencies while introducing a novel High-dimensional Position Embedding (HiPE) to encode geographic biases and reveal latent relationships. Empirical results demonstrate that POET achieves consistent state-of-the-art performance across weather, flood, and air-quality forecasting tasks at both global and regional scales.

**Strengths:**

1. The paper is clearly written with a well-structured presentation of the methodology and experimental findings.
2. The proposed HiPE effectively integrating prior knowledge and data-driven positional information.
3. The three-level attention mechanism allows POET to model intricate dependencies across different dimensions.
4. Extensive experiments across diverse Earth system forecasting tasks provide strong evidence of the model’s effectiveness.

**Weaknesses:**

* For the positional embedding design, the paper should compare POET’s HiPE with other widely used baselines, such as learnable node-specific embeddings [1] or fixed geographic encodings (e.g., DIRECT, CARTESIAN3D, and WRAP as discussed in the appendix of [2]). Such comparisons would strengthen the justification for HiPE and more clearly demonstrate its advantages over existing positional encoding methods in Earth system forecasting.
[1] Taming Local Effects in Graph-based Spatiotemporal Forecasting
[2] Geographic Location Encoding with Spherical Harmonics and Sinusoidal Representation Networks

* The attention mechanism across three dimensions is computationally expensive for large numbers of stations. Currently, POET is feasiably evaluated on datasets with 3,850 stations. For global weather forecasting, the station numbers are generally more than 10,000, how to accelerate model inference?

* POET disentangles spatial and temporal modeling. Whether it is possible to model joint spatial-temporal attention, which is demonstrated effectiveness in other tasks (e.g., ViViT and Sora)?

* From Table 3, the performance improvement for regional dataset (e.g., Flood, Germany) is relatively small.

**Questions:**

* Additional positional embedding baselines.
* Computational efficiency discussion.
* Additional insights on model designs.

**Details Of Ethics Concerns:**

N/A.

---

> ### Author Response · Authors · 2025-11-24
> **Response to Reviewer 2Zqh**
>
> Many thanks to Reviewer 2Zqh for providing a valuable review.
>
> > **W1&Q1:** For the positional embedding design, the paper should compare POET’s HiPE with other widely used baselines, such as learnable node-specific embeddings [1] or fixed geographic encodings (e.g., DIRECT, CARTESIAN3D, and WRAP).
>
> Thanks for your valuable suggestion. Our proposed HiPE comprises both absolute prior position and learnable position to uncover hidden relationships within the observed data that may not be explicitly described by existing information.
>
> Following your suggestion, we have newly included several node embedding and geographic encoding works as baselines for comparison. As listed in the Table below, compared to not using any Position Embedding method (w/o PE), the inclusion of position encoding enhances forecasting performance, with POET consistently achieving superior results across all forecasting horizons. We have included the experimental results in $\underline{\text{Table 4 of the revision}}$.
>
> |Method|w/o PE |Node Embed[1]|DIRECT[2]|CARTESIAN3D[2]|WRAP[2]|SH[2]|POET|
> |:-------: |:-------: |:-------: |:-------: |:-------: |:-------: |:-------: |:-------: |
> | 24H   |25.993|24.131   | 24.316  | 24.197 | 24.164 |25.083|23.19|
> | 48H   |32.325|30.619   | 30.823  | 31.011 | 30.620 |31.775|29.302|
> | 72H   |36.988|35.054   | 35.657  | 35.845 | 35.203 |36.469|33.530|
> | AVG  |**31.768**|**29.935**|**30.265**|**30.351**|**29.996**|**31.109**|**28.675**|
>
> > **W2&Q2:** The attention mechanism across three dimensions is computationally expensive for large numbers of stations. Currently, POET is feasiably evaluated on datasets with 3,850 stations. For global weather forecasting, the station numbers are generally more than 10,000. How to accelerate model inference?
>
> Thank you for raising this insightful question. We agree that modeling a large number of stations (e.g., over 10,000) poses computational challenges due to the quadratic complexity of the standard attention mechanism.  However, recent advancements in efficient attention mechanisms, which are widely adopted in modern large language models (LLMs), provide practical solutions to this issue. Methods such as Flash Attention [1] reduce the computational and memory overhead of the attention mechanism, enabling Transformers to handle much larger token sequences while maintaining the original accuracy. To validate this, we performed experiments on a single NVIDIA A100 Tensor Core GPU. As shown in the table below, **incorporating Flash Attention significantly reduces memory usage and makes modeling 10,000 sites feasible**.
>
> |Method|Memory Usage|
> |:-------: |:-------: |
> |3850 stations w/o Flash Attn|12468 MB|
> |3850 stations w/ Flash Attn|5782 MB|
> |10000 stations w/ Flash Attn|13196 MB|
>
> [1] Dao, T., et al. "FlashAttention: Fast and Memory-Efficient Exact Attention with IO-Awareness." NeurIPS 2022.
>
> > **W3&Q3:** POET disentangles spatial and temporal modeling. Whether it is possible to model joint spatial-temporal attention, which is demonstrated effectiveness in other tasks (e.g., ViViT and Sora)?
>
> Thank you for your valuable suggestion. In response to your request, we have **added a new baseline based on joint spatial-temporal attention** following [1]. Specifically, we adopt the spatial-temporal attention mechanism proposed in ViViT, where each transformer layer models all pairwise interactions among spatial-temporal tokens, thereby capturing long-range dependencies across all stations over time. Additionally, we integrate our proposed HiPE into the joint modeling process by stacking the three-dimensional positional embeddings into a higher-dimensional position. The results demonstrate that POET outperforms the joint spatial-temporal attention model. Moreover, HiPE can be incorporated into the joint attention mechanism, leading to further performance improvements.
>
> |Method|POET      |Joint     |Joint+HiPE|
> |:-------: |:-------: |:-------: |:-------: |
> | 24H   |23.194    | 25.860   | 24.880  |
> | 48H   |29.302    | 32.298   | 31.357  |
> | 72H   |33.530    | 36.818   | 35.473  |
> | AVG  |**28.675**|**31.659**|**30.570**|
>
> [1] Arnab, Anurag, et al. "Vivit: A video vision transformer." ICCV 2021.
>
> > **W4:** From Table 3, the performance improvement for regional dataset (e.g., Flood, Germany) is relatively small.
>
> We acknowledge that the performance gains are relatively marginal in these cases, which is likely because stations are clustered in a small area, so spatial variance is low and positional priors add little extra information for modeling inter-series correlations.
>
> However, we would like to highlight that, for global datasets such as Wind and Temp, where data points span diverse and distant geographical locations, HiPE effectively leverages prior positional information to capture spatial relationships, resulting in more significant performance improvements.

---

### Author Response · Authors · 2025-11-24
**Summary of Revisions**

We sincerely thank all the reviewers for their valuable feedback, which is instructive for us to improve our paper.

**Our contribution:** In this paper, we dive into high-dimensional Earth system observations and introduce POET as a novel Transformer-based architecture for Earth system forecasting, which organizes and effectively captures the dependencies within the Earth system in three dimensions: time, space, and variate. Moreover, POET employs a High-dimensional Positional Embedding mechanism (HiPE) to utilize both learned and prior position information to guide the attention learning. Experimentally, POET consistently achieves state-of-the-art performance with favorable interpretability in diverse real-world benchmarks.

We are pleased that all the reviewers appreciate the novelty and contribution of POET:

- **2Zqh**: "The proposed HiPE **effectively integrates** prior knowledge and data-driven positional information.", and the experiments "**provide strong evidence** of the model’s effectiveness".
- **UAmW**: "The paper **tackles a challenge that is common in many Earth modelling tasks**", "The experiments are sound".
- **pGZZ**: "The article is **well written** and the figures are of **high quality**; it is easy to follow." "The model can **generalize well** to a wide range of applications ", "Integrating the positional encoding in a layer is **smart**", and "utilizing RoPE instead of a relative positional encoding seems **quite promising**".
- **AXCp**: "**has many concrete applications and direct pathways for impacting real-world modeling and decision making**", "The proposed neural net architecture is **explained well**".

**Summary of Rebuttal:** The reviewers also raised insightful and constructive concerns. We further improved this work in the following aspects. All the discussion and experiments have been included in the $\underline{\text{revised paper}}$.

- **Additional baselines on location encoding** (Reviewer 2Zqh, UAmW, pGZZ, AXCp): Following the reviewers' suggestions, we conducted comprehensive comparisons between the proposed HiPE and **twelve advanced spatial embedding methods**.

- **Analyze the model performance under challenging settings** (Reviewer 2Zqh, UAmW): As per the reviewer's request, we evaluated the model under challenging conditions, including settings with a large number of stations and with missing observations, where HiPE demonstrates its effectiveness and robustness under practical scenarios.

- **Elaborate on the effectiveness of HiPE** (Reviewer UAmW, AXCp): We provided in-depth explanations regarding the use of raw coordinates. We highlight that the visualization of learned positions reveals the underlying relations inherent in the data, which can provide valuable insights that enhance data understanding.

- **Elaborate on the problem POET tackles** (Reviewer UAmW, pGZZ): We clarified that POET targets a common challenge in Earth modeling where observations are collected from irregularly distributed stations. These stations may be globally or locally distributed, and POET demonstrates superior performance in both scenarios.

- **Polished presentation** (Reviewer pGZZ, AXCp): We have conducted detailed proofreading and revisions with helpful suggestions from the reviewers, including the term knowledge position and format of Table 2.

We deeply thank all the reviewers for their effort in reviewing our paper, which helps us a lot in improving our work's experiments and scientific rigor. We'd be very happy to answer any further questions.

Looking forward to the reviewer's feedback.

---

### Meta-Review · Area_Chair_6uLb · 2025-12-24

**Summary:**

This paper studies Earth system forecasting under irregular and partially observed station settings and proposes the Partially Observed Earth Transformer (POET), together with a High-Dimensional Position Embedding (HiPE) strategy to incorporate spatial, temporal, and variate positional information into attention. Reviewers generally agree that the problem setting is important, the model is carefully engineered, and the empirical results across weather, air quality, and flood forecasting benchmarks are strong. The rebuttal provides extensive additional experiments, including comparisons with location encoding baselines, joint spatio-temporal attention, and robustness under missing data. However, after considering the rebuttal and post-discussion feedback, substantial concerns remain regarding the clarity of the core methodological motivation, the necessity and conceptual distinctiveness of the proposed architecture over existing spatio-temporal and location encoding approaches, and the alignment between the claimed “partially observed” setting and the evaluated problem formulation. In particular, while HiPE demonstrates empirical effectiveness, its contribution appears largely incremental and engineering-driven, with limited theoretical or principled justification for why the proposed design is fundamentally required. While the work is solid, well executed, and practically relevant, the remaining concerns regarding methodological motivation and conceptual distinctiveness make it a borderline case with respect to the level of methodological novelty typically expected for publication at ICLR.

**Reviewer Concerns:**

Reviewer 2Zqh: The rebuttal adds extensive baselines and scalability analysis, but concerns about the necessity of the cascaded attention design and the incremental nature of HiPE over existing positional encoding methods remain.

Reviewer pGZZ: Additional comparisons and ablations strengthen the empirical results, yet the core novelty relative to prior spatio-temporal Transformer models is still not fully convincing.

Reviewer UAmW: Despite the added experiments and clarifications, the motivation of the proposed architecture and the distinction from existing location embedding and space-time encoding methods remain insufficiently clear.

Reviewer AXCp: The work is well written and practically relevant, but the methodological contribution appears more incremental than conceptual, limiting its impact at a top-tier venue.

**Reviewer Scores:**

Reviewer 2Zqh: Likely no change.

Reviewer pGZZ: Likely no change.

Reviewer UAmW: Likely no change.

Reviewer AXCp: Likely no change or a slight increase.

---

### Decision · Program_Chairs · 2026-01-26

Reject